# Cytokinesis in vertebrate cells initiates by contraction of an equatorial actomyosin network composed of randomly oriented filaments

**Felix Spira[1], Sara Cuylen-Haering[1†], Shalin Mehta[2‡], Matthias Samwer[1], Anne Reversat[3], Amitabh Verma[2], Rudolf Oldenbourg[2], Michael Sixt[3], Daniel W Gerlich[1]***

[1]Institute of Molecular Biotechnology of the Austrian Academy of Sciences, Vienna Biocenter, Vienna, Austria; [2]Eugene Bell Center for Regenerative Biology and Tissue Engineering, Marine Biological Laboratory, Woods Hole, United States; [3]Institute of Science and Technology Austria, Klosterneuburg, Austria

**Abstract** The actomyosin ring generates force to ingress the cytokinetic cleavage furrow in animal cells, yet its filament organization and the mechanism of contractility is not well understood. We quantified actin filament order in human cells using fluorescence polarization microscopy and found that cleavage furrow ingression initiates by contraction of an equatorial actin network with randomly oriented filaments. The network subsequently gradually reoriented actin filaments along the cell equator. This strictly depended on myosin II activity, suggesting local network reorganization by mechanical forces. Cortical laser microsurgery revealed that during cytokinesis progression, mechanical tension increased substantially along the direction of the cell equator, while the network contracted laterally along the pole-to-pole axis without a detectable increase in tension. Our data suggest that an asymmetric increase in cortical tension promotes filament reorientation along the cytokinetic cleavage furrow, which might have implications for diverse other biological processes involving actomyosin rings.

DOI: https://doi.org/10.7554/eLife.30867.001

***For correspondence:**
Daniel.Gerlich@imba.oeaw.ac.at

**Present address:** [†]European Molecular Biology Laboratory Heidelberg, Heidelberg, Germany; [‡]Chan Zuckerberg Biohub, San Francisco, United States

**Competing interests:** The authors declare that no competing interests exist.

## Introduction

Animal cells divide by contracting the cell cortex in an equatorial zone between the two poles of the mitotic spindle, resulting in the ingression of a cleavage furrow. This is initiated by activation of the small GTPase RhoA at the equatorial cell cortex, which induces polymerization of unbranched actin filaments and activation of non-muscle myosin II (*Piekny et al., 2005*; *Bement et al., 2006*). The equatorial zone of actin and myosin enrichment, termed the actomyosin ring, was discovered more than four decades ago (*Schroeder, 1968*; *Schroeder, 1972*), yet how its filaments organize to generate contractile force remains poorly understood (*Eggert et al., 2006*; *Green et al., 2012*; *Fededa and Gerlich, 2012*; *Cheffings et al., 2016*; *Wang, 2005*).

Electron microscopy revealed that many actin filaments orient along the edge of the cytokinetic cleavage furrow in marine embryos (*Schroeder, 1972*; *Henson et al., 2017*) and vertebrate cells (*Maupin and Pollard, 1986*). This led to the suggestion that the actomyosin ring might contract by anti-parallel filament sliding similar to muscle sarcomeres (*Schroeder, 1972*). The actin filament alignment during the initiation of furrow ingression, however, appears less prominent (*Mabuchi, 1994*; *Fishkind and Wang, 1993*; *Noguchi and Mabuchi, 2001*; *DeBiasio et al., 1996*; *Fenix et al., 2016*) and whether the filament sliding model appropriately describes animal cell

cytokinesis has remained under debate (*Eggert et al., 2006*; *Green et al., 2012*; *Fededa and Gerlich, 2012*; *Cheffings et al., 2016*; *Wang, 2005*).

In vitro experiments showed that disorganized actomyosin networks can also generate contractile force (*Pollard, 1976*; *Janson et al., 1991*; *Murrell et al., 2015*; *Linsmeier et al., 2016*; *Ennomani et al., 2016*), which is explained by the imbalance of tensile over compressive forces that can be transduced by actin filaments (*Murrell et al., 2015*; *Linsmeier et al., 2016*; *Murrell and Gardel, 2012*; *Vogel et al., 2013*; *Lenz et al., 2012*; *Lenz, 2014*). It is therefore possible that the actomyosin ring ingresses by network contraction rather than sliding of anti-parallel filament bundles.

Theoretical models suggest that actin filament alignment during cytokinesis might be a consequence rather than cause of actomyosin contractility (*Greenspan, 1977*; *White and Borisy, 1983*; *Bray and White, 1988*; *Zinemanas and Nir, 1988*; *Verkhovsky et al., 1999*; *Salbreux et al., 2009*; *Stachowiak et al., 2014*; *Dorn et al., 2016*; *Reymann et al., 2016*). In large cells of *Caenorhabditis elegans* early embryos, actin filament alignment is driven by a myosin-dependent long-range cortical flow that precedes furrow ingression (*Reymann et al., 2016*). In fission yeast, actin filaments nucleate at discrete loci that interact and contract to align filaments at the cell equator (*Stachowiak et al., 2014*; *Wu et al., 2006*; *Vavylonis et al., 2008*; *Kamasaki et al., 2007*; *Laplante et al., 2016*). Dividing vertebrate cells do not have prominent actin nucleation foci and they have less pronounced cortical flow compared to large embryo cells (*Cao and Wang, 1990*; *Murthy and Wadsworth, 2005*; *Zhou and Wang, 2008*). It has thus remained unclear how myosin-mediated contractility contributes to filament organization within the actomyosin ring of vertebrate cells.

The direct visualization of filament orientation within the actomyosin ring is difficult owing to the resolution limit of light microscopes and potential fixation and staining artifacts in electron microscopy. Fluorescence polarization microscopy, however, provides an alternative approach to quantify filament orientation independently of the resolution limit of the microscope, by reporting the net orientation of fluorophore dipoles bound to filaments (*DeMay et al., 2011a*; *McQuilken et al., 2015*; *Mehta et al., 2016*). Fluorescence polarization microscopy indeed revealed septin filament reorientations in the bud neck of *Saccharomyces cerevisiae* (*DeMay et al., 2011b*; *Vrabioiu and Mitchison, 2006*), actin filament anisotropy during cytokinesis of vertebrate cells (*Fishkind and Wang, 1993*) and *Drosophila* embryos, (*Mavrakis et al., 2014*) and dynamic orientation of actin filaments within the lamellipodium of migrating cells (*Mehta et al., 2016*). While prior fluorescence polarization microscopy analysis showed an increase of actin filament alignment at late stages of cytokinesis in mammalian cells (*Fishkind and Wang, 1993*), the precise timing and absolute degree of actomyosin network order during cytokinesis, the distribution of forces during cytokinesis progression, and the mechanisms underlying actin filament alignment have remained unclear.

Here, we studied the kinetics of actin filament reorganization during cytokinesis in human cells based on calibrated fluorescence polarization microscopy, using fixed- and live-cell probes. We further measured the cortical tension generated by the actomyosin cortex during different stages of cytokinesis, using laser microsurgery. Our data indicate that cytokinetic cleavage furrow ingression initiates by contraction of a equatorial actomyosin network composed of randomly oriented filaments. The network then partially aligns its filaments by myosin-driven contraction toward the equator, thereby further increasing tension along the cleavage furrow.

## Results

### Contraction of a randomly oriented actin network initiates cleavage furrow ingression

Actin filaments are known to align with the cleavage furrow during late stages of cytokinesis. Yet, it has remained unclear whether actin filament alignment is required for initiation of furrow ingression, as proposed by the canonical 'purse-string' model of contractility. To determine the degree of actin filament order in the cortex of dividing cells, we visualized actin filament orientation with phalloidin staining and a custom fluorescence polarization confocal microscopy setup based on a liquid crystal universal compensator (*DeMay et al., 2011a*) (LC-PolScope). Alexa488-phalloidin was previously shown to bind to F-actin filaments with defined orientation of the fluorophore dipoles (*Mehta et al., 2016*). The LC-PolScope enables to analyze the orientation of these fluorophores and therefore the

orientation of the actin filaments. To this end, the liquid crystal modulates the linear polarization direction of the excitation laser into four different angles (*Figure 1—figure supplement 1A*). We calibrated the imaging set-up by using samples showing maximal and minimal fluorophore anisotropy and used these calibration measurements to calculate a 'normalized polarization factor'.

Maximally anisotropic fluorophore orientation is expected for parallel filaments that are aligned with the optical section (*Figure 1A* 'parallel view'). We therefore imaged bundles of aligned actin filaments in stress fibers at the bottom surface of interphase human diploid Retinal Pigment Epithelium (hTERT-RPE-1) cells using the confocal LC-PolScope (*Figure 1B*). As a sample with isotropic fluorescence dipole distribution, we imaged fluorescent plastic (*Figure 1C*). We then normalized the polarization factor to be in a range between 1 and 0 for maximally aligned fluorophores in stress fibers and randomly oriented fluorophores in plastic, respectively. Together, these data provide a reference scale to quantify fluorophore polarization (*Figure 1C*) and therefore actin filament orientation.

We next investigated actin filament organization in mitotic cells. The actin cortex underlying the plasma membrane of mitotic cells is relatively thin (*Chugh et al., 2017*), which might confine actin filaments into a flat network. If actin filaments in the mitotic cell cortex were oriented randomly in the plane along the cell surface, then a confocal optical section of the LC-PolScope through the center of a metaphase cell should capture a mixed population of actin filaments with angles ranging in between parallel to the cell surface and optical plane (*Figure 1A* 'parallel view') and perpendicular to the optical plane (*Figure 1A* 'perpendicular view'). As each camera image pixel records signal from many fluorophores, this should yield a normalized polarization factor close to 0.5 along the metaphase cell cortex. To test this, we recorded central optical sections of Alexa488-phalloidin-stained metaphase cells, segmented the cortex by using an image contour that excluded filopodia (*Figure 1—figure supplement 1B,C*), measured the fluorophore anisotropy and normalized it to the reference scale described above. The detected normalized polarization factor of 0.57 ± 0.09 (*Figure 1C*; median ±s.d., as for all following measurements) was indeed close to the value expected for a thin isotropic actin network in metaphase cells, indicating that the confocal LC-PolScope is suitable for quantitative analysis of degree of filament order within actin networks.

We then investigated how the cortical actin network reorganizes when cells proceed from metaphase to anaphase. The canonical 'purse-string' model of cytokinesis (*Schroeder, 1972*) proposes that actin filaments align along the cell equator to drive furrow ingression by anti-parallel sliding (*Figure 1D*). As these filaments would orient perpendicular to the confocal optical section, this should result in a decrease of the normalized polarization factor at the equator, while the randomly oriented network at the poles should maintain values similar to metaphase cells (*Figure 1D*). To test this, we quantified the orientation of cortical actin filaments during progression from metaphase into anaphase and furrow ingression. We recorded polarized fluorescence images of fixed hTERT-RPE-1 cells stained with Alexa488-phalloidin and 4′,6-Diamidin-2-phenylindol (DAPI) as a reference DNA dye, and binned cells according to cell cycle progression based on cleavage furrow diameter and chromosome-chromosome distance (*Figure 1E* and *Figure 1—figure supplement 2A and B*). At late stages of cytokinesis, we indeed observed a pronounced reduction of the normalized polarization factor at the cell equator (*Figures 1E*, 201–300 s, cortex area with low-color saturation; quantification in 1F), consistent with preferential orientation of actin filaments along the cell equator. The reduction of the normalized polarization factor at the cell equator cannot be attributed to the higher levels of mean Alexa488-phalloidin fluorescence, as fluctuations of mean fluorescence along the cell cortex did not correlate with the degree of polarization at early cytokinesis (*Figure 1—figure supplement 1C*). Our setup can therefore detect the equatorial actin network reorganization that was previously described during late stages of cytokinesis (*Fishkind and Wang, 1993*).

We then asked how cortical actin network orientation correlates with changes of cell geometry during anaphase. Following the onset of chromosome segregation in an almost spherical cell, anaphase cells start to ingress a cleavage furrow after 101–150 s, visible as a curvature reduction at the equator (*Figure 1E*). At these early furrow ingression states, the normalized polarization factor was indistinguishable between equator and pole (*Figure 1G*). Only at later furrow ingression states (151–300 s), the normalized polarization factor at the cleavage furrow decreased significantly compared to the pole and the initial metaphase values (*Figure 1G*). We also noticed a small increase of normalized polarization factor at the poles during late stages of cytokinesis, which might reflect cortical reorganization when cells reattach to the glass substratum during mitotic exit. The polarization

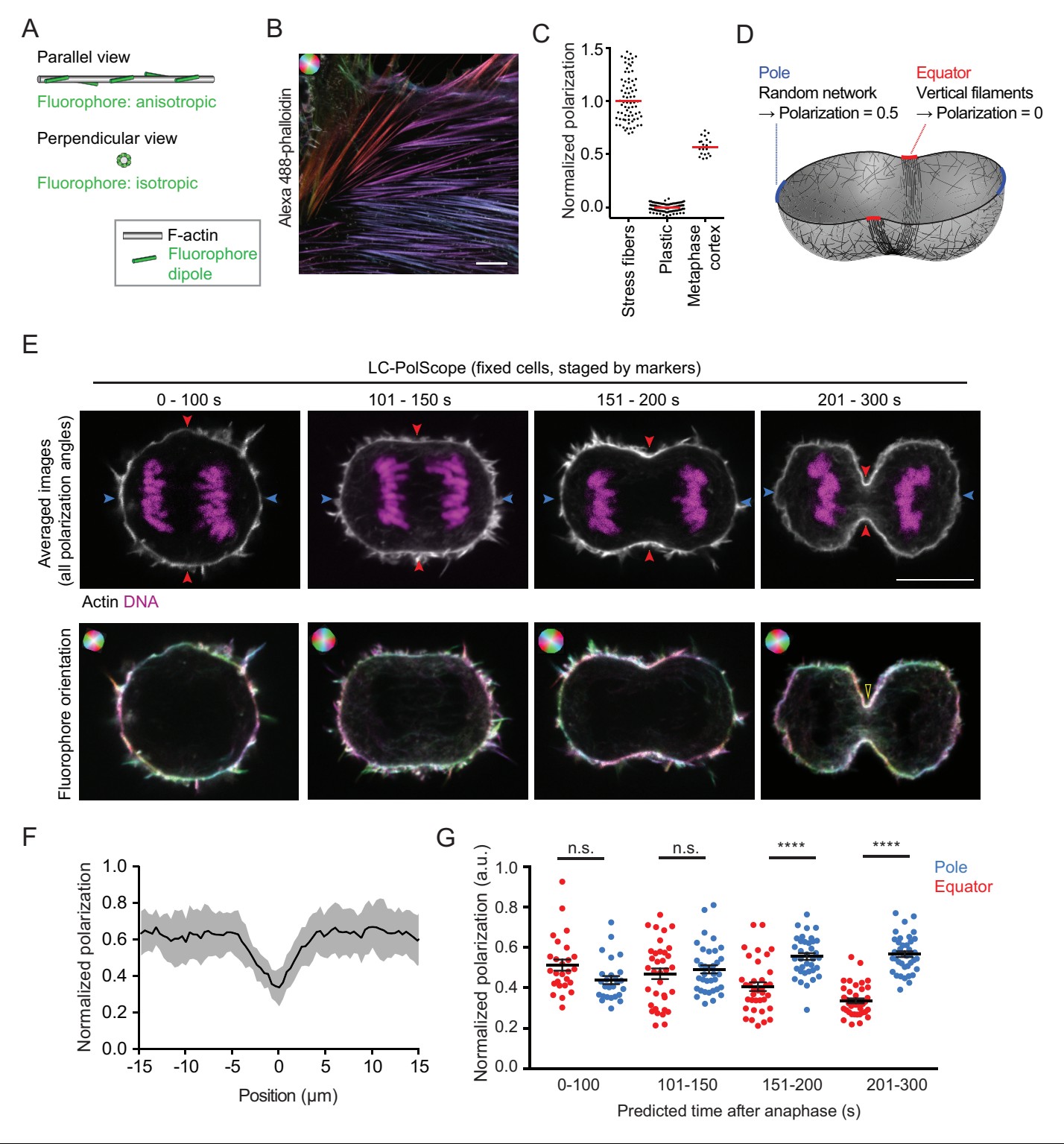

**Figure 1.** Cleavage furrow ingression initiates by contraction of a randomly oriented actin filament network, which subsequently gradually aligns at the cell equator. (**A**) Schematics of actin filament (grey) and fluorescent dipole orientation (green) relative to the optical section of the microscope. Upper panel shows actin filament parallel to the focal plane of the microscope, lower panel shows an actin filament that is perpendicularly oriented to the focal plane of the microscope. On perpendicularly oriented actin filaments, fluorescent dipoles are oriented point symmetrically in every direction and the ensemble of molecules therefore does not yield a fluorescence anisotropy signal. (**B**) Fluorescence polarization microscopy of stress fibers in fixed interphase hTERT-RPE-1 cells labeled with Alexa Fluor 488-phalloidin using the LC-PolScope. Color saturation indicates degree of fluorophore

*Figure 1 continued on next page*

*Figure 1 continued*

alignment (anisotropy), hue indicates mean orientation of fluorescence dipoles as shown in upper left corner. (C) Quantification of polarization normalized to calibration samples: stress fibers as in (B) and fluorescent plastic with random fluorescence dipole orientation. Dots indicate individual measurements, bars indicate median. (D) Geometry and normalized polarization predicted by canonical purse-string model of cytokinesis at the equator (red) or cell poles (blue). (E) Images of Alexa Fluor 488-phalloidin-stained hTERT-RPE-1 cells at representative stages during cytokinesis. Blue and red arrows indicate polar and equatorial positions of quantification regions, respectively (upper panel). Lower panel shows the orientation map of the fluorescent dipole as calculated by the different orientations. Color saturation indicates degree of fluorophore alignment (anisotropy), hue indicates mean orientation of fluorescence dipoles as shown in upper left corner. Yellow arrowhead indicates edge of cleavage furrow. (F) Lateral distribution of polarization factor measured along the cell cortex in central sections of late furrow ingression-stage cells, aligned for the cleavage furrow edge as in (E). Line indicates median, shaded area indicates s.d. of 22 cells. (G) Quantification of normalized polarization as indicated in (E). Dots indicate individual cells (n = 136, mean + s.e.m. of both measurements at opposing cortical cell regions; ****p<0.0001 by Kolmogorov-Smirnov test). Scale bars = 10 μm.

DOI: https://doi.org/10.7554/eLife.30867.002

The following figure supplements are available for figure 1:

**Figure supplement 1.** LC-PolScope fluorescence polarization microscopy and analysis pipeline.

DOI: https://doi.org/10.7554/eLife.30867.003

**Figure supplement 2.** Regression model to determine cytokinesis timing in fixed cells.

DOI: https://doi.org/10.7554/eLife.30867.004

measurements varied quite substantially between individual cells, yet owing to the relatively large sample sizes, we obtained accurate estimates of the mean for each of the respective stages and cellular positions (s.e.m. ranging between 0.01 and 0.03, *Figure 1G*). Overall, these observations indicate that the orientation of actin filaments in the equatorial network driving initial stages of constriction is indistinguishable from the random orientation observed in the cortex of metaphase cells.

We next aimed to resolve the dynamics of actin network reorganization during cytokinesis with higher temporal resolution in living cells. Fluorescent phalloidin cannot be used for this purpose, as it does not passage the plasma membrane and is known to be highly toxic. We thus tested a silicone rhodamine-coupled jasplakinolide derivate, SiR-actin, which is a plasma membrane-permeant, fluorogenic actin marker that has low toxicity (*Lukinavičius et al., 2014*). SiR-actin indeed generated a strong polarization-dependent signal on stress fibers of live hTERT-RPE-1 cells (*Figure 2A*), and it did not impair the efficiency and rate of cleavage furrow ingression at the concentration used for imaging (*Figure 2—figure supplement 2B and C*). We therefore used SiR-actin to probe actin filament orientations in live cells progressing through cytokinesis.

The relatively slow image acquisition rate with the LC-PolScope (~20 s per frame) leads to motion artifacts when imaging fast-moving structures like the cytokinetic cleavage furrow. We hence used a complementary microscopy setup for live-cell analysis of cytokinesis, with two perpendicularly oriented linear polarization filters positioned in the fluorescence emission light path (*Figure 2—figure supplement 1A*, *Figure 2B*). This live-cell setup yields a fluorophore emission ratio image that reports on relative changes of fluorophore anisotropy over time. We normalized the fluorophore emission ratio to metaphase as a reference time point (normalized emission ratio) and then studied relative changes during anaphase progression.

The live-cell setup uses only two perpendicular polarization filter angles to calculate the fluorescence emission ratio, which precludes the detection of anisotropy at angles close to 45° and 135° relative to the axes of the polarization filters. Furthermore, fluorescence emission ratio measurements are affected by cell geometry. This is illustrated by images of metaphase cortices, which show that the fluorophore emission ratio along the cell cortex follows a sine squared function (*Figure 2—figure supplements 1D,E*). We implemented a procedure to correct for such geometry effects (*Figure 2—figure supplement 1F*, see Materials and methods). To minimize the effect of cell geometries, we imaged cells that had their spindle axis aligned with the optical table (90°±20° relative to the X-axis of the optical table) and considered only time points in which the absolute furrow cortex curvature did not exceed that of metaphase cells. For this, we measured cortex curvature by fitting circles to the equatorial cell cortex (*Figure 2C*). We then determined the time window in which the absolute value of inward curvature was not higher than the absolute value of outward curvature at metaphase (*Figure 2C* 'metaphase' – 'early ingression' and *Figure 2D–F*, green areas). We

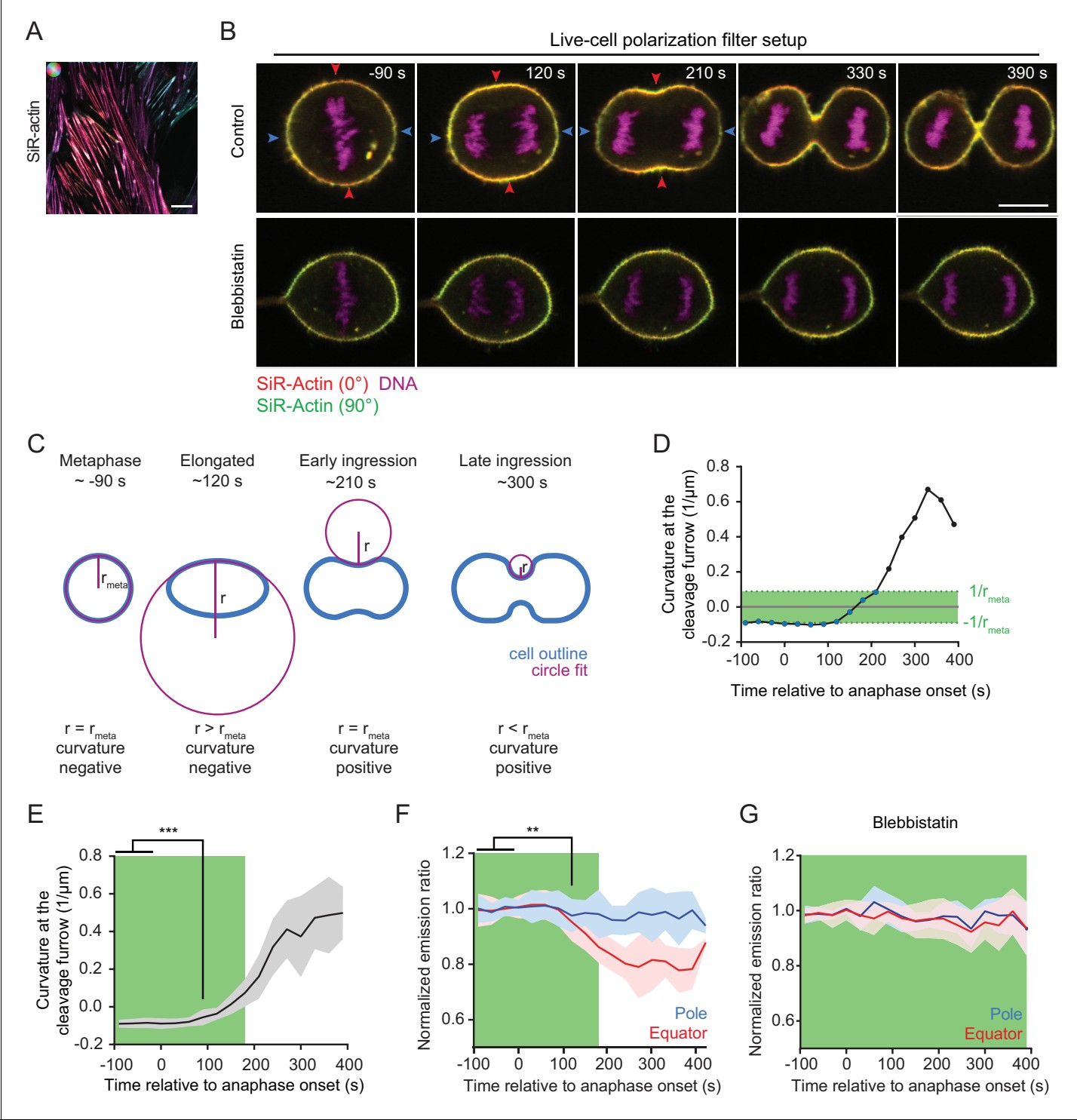

**Figure 2.** Quantification of actin reorganization by live-cell confocal polarization microscopy. (**A**) Fluorescence polarization microscopy of stress fibers in live interphase hTERT-RPE-1 cells labeled with SiR-actin using the LC-PolScope. Color saturation indicates degree of fluorophore alignment (anisotropy), hue indicates mean orientation of fluorescence dipoles as shown in upper left corner. (**B**) Fluorescence polarization microscopy of live hTERT-RPE-1 cells stably expressing H2B-mRFP (magenta), stained with SiR-actin (overlay of red and green) at representative stages during cytokinesis. Images were acquired with horizontal (red) and vertical (green) linear polarizers in the emission beam path. Untreated cells (upper panel) and cells treated with 50 µm para-nitroblebbistatin (lower panel) are shown. (**C**) Schematic drawing of the circular fit procedure during cell division. Circles (magenta) were fitted to the center of the equator (blue). (**D**) Curvature of the equatorial cell cortex in a representative cell. Dots indicate individual time points. Time = 0 s at anaphase onset. (**E**) Quantification of equatorial cortex curvature in 18 dividing cells. Line indicates mean, gray area s.d.. The

*Figure 2 continued on next page*

*Figure 2 continued*

first time point with significantly changed curvature was detected at 90 s after anaphase onset by two-tailed t-test; ***p<0.001. (F) Quantification of SiR-actin normalized emission ratio in the 18 dividing cells used for the analysis in (E) (line indicates mean, blue and red areas s.d. **p<0.01 by two-tailed t-test (G) Quantification of SiR-actin normalized emission ratio in dividing cells treated with 50 µm para-nitroblebbistatin. Time = 0 s at anaphase onset (lines indicate median, shaded areas indicate s.d.; equatorial measurement: n = 15 cells, polar: n = 11 cells). (D-G) The green area indicates time points where the absolute value of cortical curvature was equal or below that of the metaphase cell. Only these time points were used for further interpretation of fluorescence anisotropy measurements, to avoid potential artifacts by geometry effects. Scale bars = 10 µm.
DOI: https://doi.org/10.7554/eLife.30867.005

The following figure supplements are available for figure 2:

**Figure supplement 1.** Polarization microscopy with linear polarizers in the emission beam path.
DOI: https://doi.org/10.7554/eLife.30867.006
**Figure supplement 2.** Analysis pipeline of live-cell polarization microscopy.
DOI: https://doi.org/10.7554/eLife.30867.007
**Figure supplement 3.** Cortex organization in para-nitroblebbistatin-treated anaphase cells and lateral distribution of actin and myosin in untreated cells.
DOI: https://doi.org/10.7554/eLife.30867.008

selected only these time points for further analysis of fluorescence anisotropy, thereby comparing states with matching cortex orientations.

We then asked whether actin filament alignment with the cleavage furrow precedes cleavage furrow ingression, or whether it occurs only after ingression onset in living cells. To determine the onset of cleavage furrow ingression, we compared the equatorial cortex curvatures at different time points of anaphase with the curvatures of the equatorial metaphase cortex. We detected significant inward-directed curvature changes at 90 s post anaphase onset (*Figure 2E*; n = 18 movies). Quantification of the equatorial emission ratio of SiR-actin in the same data set revealed significant changes only at 120 s after anaphase onset (*Figure 2F* and *Figure 2—figure supplement 2A and B*). Thus, furrow initiation precedes detectable changes in equatorial actin network orientation in vertebrate cells.

Our normalized emission ratio measurements during cytokinesis progression might be affected by changing thickness of the actin cortex. We therefore measured the thickness of the SiR-actin-labeled cell cortex, using a plasma membrane marker as reference (*Figure 2—figure supplement 1G–J*). We found that the actin cortex in metaphase cells is thinner than in interphase cells, consistent with prior findings (*Chugh et al., 2017*). Importantly, however, we did not detect a significant difference between the thickness of the equatorial cortex of metaphase and anaphase cells (*Figure 2—figure supplement 1J*), indicating that equatorial actin accumulations in anaphase cells must be due to increased local filament density. The equatorial fluorescence anisotropy changes detected during cytokinesis progression hence cannot be attributed to cortical thickening.

## Equatorial actin filament reorientation depends on myosin II activity

Actin filament alignment at the cell equator might result from spatially confined polymerization or from motor-driven filament reorientation. To investigate this, we tested whether myosin activity is required to reorient equatorial actin filaments during anaphase. Blebbistatin is a specific inhibitor of non-muscle myosin II that suppresses cleavage furrow ingression but does not prevent accumulation of actin and myosin II at the equatorial cell cortex (*Zhou and Wang, 2008*; *Straight et al., 2003*). Para-nitroblebbistatin is a more photostable version of this inhibitor (*Képiró et al., 2014*), and it completely suppressed cleavage furrow ingression in hTERT-RPE-1 cells (*Figure 2B*, 'Blebbistatin' and *Figure 2—figure supplement 3A*). Actin filament accumulation at the cell equator was not affected in blebbistatin-treated cells (*Figure 2—figure supplement 3B and C*), yet emission ratio changes of SiR-actin at the equator were completely suppressed. (*Figure 2G*). Thus, myosin II activity is not required for equatorial actin filament accumulation but is essential for filament reorientation along the cell circumference.

Given that equatorial actin network reorientation depended on myosin II activity, we investigated whether the local concentration of myosin II correlates with the degree of actin filament alignment. To test this, we imaged live hTERT-RPE-1 cells stably expressing regulatory myosin light chain 12B tagged with EGFP (MLC-12B-EGFP), counterstained with SiR-actin. Myosin II enriched at the cell

cortex along a gradient that peaked at the cleavage furrow, with a full width at half maximum of 4.3 ± 0.9 μm (*Figure 2—figure supplement 3D and E*). This distribution closely matched the lateral distribution of anisotropy within the actin network (4.4 ± 1.5 μm, see *Figure 1F*). The peak of SiR-actin fluorescence also localized to the cell equator, but the distribution was substantially broader, with a full width at half maximum of 8.4 ± 1.8 μm (*Figure 2—figure supplement 3D and E*). This is consistent with small-scale organization within the actomyosin ring, where pronounced myosin II clusters also do not locally enrich actin to the same extent (*Wollrab et al., 2016*). Together, our data indicate that equatorial accumulation and reorientation of actin filaments within the cortex are distinctly regulated, supporting a model of myosin-mediated actin network reorganization.

## Equatorial actin filaments reorient on planar cortex regions

To study the mechanism underlying initial cleavage furrow ingression in more detail, we aimed to visualize the rearranging actin network in a single, high-resolution optical plane. A key limitation is the curved geometry of the cleavage furrow, which leads to detachment of the cell cortex from the supporting coverslip (*Figure 3—figure supplement 1A and B*). Owing to the relatively fast furrow ingression and the inherently low optical resolution of confocal microscopes along the z-axis, it is difficult to reconstruct the network rearrangements from 3-D time-lapse microscopy data. We therefore confined cells in microchambers (*Liu et al., 2015*) to locally suppress cortex detachment during anaphase (*Figure 3A*). Confined cells formed a cleavage furrow in the x-y optical plane (*Figure 3B–D*) but were substantially delayed toward ingression along the z-axis (*Figure 3D*). We next imaged live cells in confinement microchambers stained with SiR-actin and measured network reorganization. Actin filaments accumulated at the cell equator at about 90 s after anaphase onset (*Figure 3E and F*), and SiR-actin normalized emission ratio measurements began to increase at about 120 s after anaphase onset (*Figure 3E and G*). Note that in this imaging setup that analyzes actin filaments parallel to the optical plane, filament alignment leads to an increase of the normalized emission ratio (see also *Figure 1A*). Our confinement setup therefore enables to analyze actin network reorganization with higher spatial resolution.

## Cortical tension is highly directional at the cell equator

We then aimed to visualize and locally manipulate the equatorial actin filaments with high spatio-temporal resolution during cytokinesis. We imaged dividing cells stably expressing LifeAct-mCherry in confinement microchambers using a total internal reflection fluorescence microscope. LifeAct-mCherry accumulated at the cell equator and formed thick actin bundles that progressively aligned with the cleavage furrow during cytokinesis progression (*Figure 4A* and *Video 1*). Concomitant with the equatorial accumulation of LifeAct-mCherry, we observed a minimal lateral displacement of cortical actin filaments toward the equator (*Figure 4B*). This argues against a strong contribution of 'cortical flow' of actin filaments in the equatorial actin enrichment and rather indicates that most actin filaments polymerize locally at the equator of somatic human cells.

We next investigated the mechanical tension generated by the actomyosin ring by local cortical laser microsurgery (*Mayer et al., 2010*). We used a pulsed high-energy laser to cut the equatorial cell cortex along a linear path of 5 μm (*Figure 4C–F*). The cortex regions adjacent to the laser cut subsequently rapidly moved outwards (*Figure 4—figure supplement 1A–E*), whereby the cortical displacement speed followed an exponential function that is characteristic for a viscoelastic response (*Wottawah et al., 2005*). Under this regime, the initial outward movement of the displaced cortex is proportional to cortical tension (*Mayer et al., 2010*). To test whether the pulsed laser indeed cut the cortex, we performed control photobleaching experiments using a conventional excitation laser. Photobleached Actin-EGFP homogeneously recovered in the bleached cortex area without lateral outward movement (*Figure 4—figure supplement 1F*), validating that the pulsed laser indeed specifically cut the actin cortex. Imaging the plasma membrane marker Myr-Palm-GFP further showed that the pulsed laser cut both the actin cortex as well the plasma membrane (*Figure 4—figure supplement 1G*). Time-lapse microscopy showed that following the rapid cortical displacement after laser cutting, most cells repaired the cell cortex and resumed furrow ingression to completion (8 out of 12 cells within 10 min; 2 cells stopped furrow ingression and 2 cells died). Overall, these experiments validate the use of intracellular laser microsurgery to measure cortical tension in mitotic hTERT-RPE1 cells.

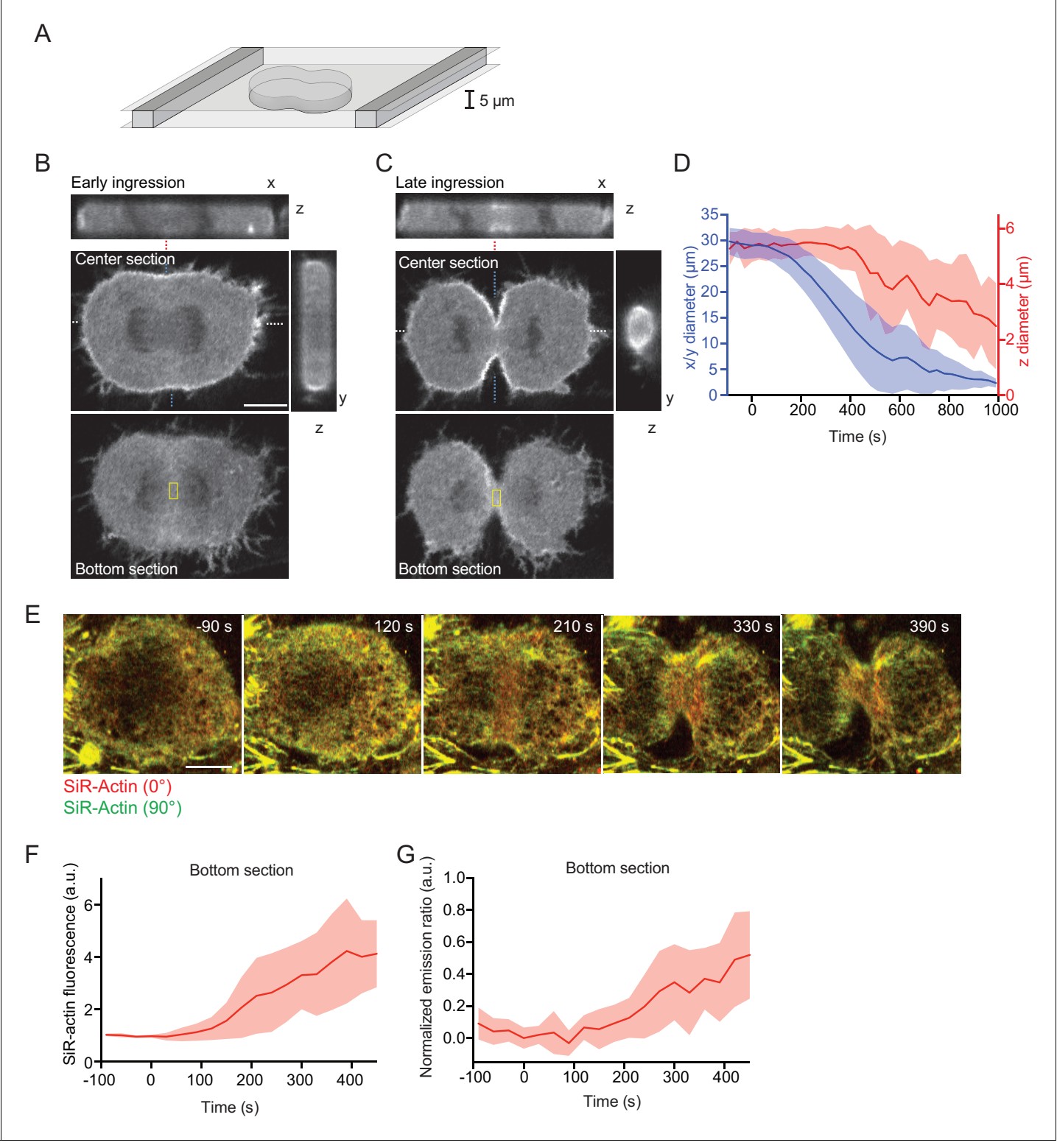

**Figure 3.** Equatorial actin filaments align on planar surfaces of confined cells. (A) Schematics of cell confinement chamber used to enforce flat geometry of top- and bottom cell cortex during cytokinesis. (B, C) Confocal 3-D images of live hTERT-RPE-1 cells stably expressing LifeAct-mCherry at (B) early or (C) late furrow ingression stages. The x/z and y/z sections are at positions as indicated by blue or white dashed lines, respectively. Dashed red lines indicate measurement region for furrow ingression along z. Yellow boxes indicate measurement regions for (F, G). (D) Quantification of cleavage furrow ingression in confined cells as in (A–C) along x/y and z directions. Lines indicate median, shaded areas s.d. of n = 14 cells. (E) Fluorescence polarization microscopy of the bottom section of live hTERT-RPE-1 cell stably expressing H2B-mRFP, stained with SiR-actin under

*Figure 3 continued on next page*

*Figure 3 continued*

confinement. Images were acquired with horizontal (red) and vertical (green) linear polarizers in the emission beam path. (**F**) Quantification of SiR-actin fluorescence intensity at the cell equator at the bottom surface of confined cells. (**G**) Quantification of SiR-actin normalized emission ratio at the cell equator at the bottom surface of confined cells. Lines indicate median, shaded areas s.d. of n = 10 cells. Scale bars = 10 μm.

DOI: https://doi.org/10.7554/eLife.30867.009

The following figure supplement is available for figure 3:

**Figure supplement 1.** 3D-imaging of cleavage furrow ingression in unconfined cells.

DOI: https://doi.org/10.7554/eLife.30867.010

We then analyzed cortical tension at the cell equator at different stages of cytokinesis. In metaphase cells, the initial outward movement, and therefore the cortical tension along the direction of the cell equator, was approximately equal to the tension along the pole-to-pole axis (*Figure 4G and H*). During early anaphase at pre-furrow ingression stages, cortical tension was similar as in metaphase, but it increased significantly along the direction of the cell equator once the cleavage furrow had begun to ingress (*Figure 4G and H*). During late furrow ingression stages, cortical tension further increased along the direction of the cell equator. Along the pole-to-pole axis, however, cortical tension remained as low as in metaphase during all stages of cytokinesis (*Figure 4G and H*). Cutting along a path oriented 45° relative to the cell equator resulted in a diagonal displacement of the cortex with most pronounced movement along direction of the cell equator (*Figure 4—figure supplement 1H*). Together, these data show that the contractility of the actomyosin ring generates cortical tension predominantly along the direction of the cell equator.

Finally, we investigated whether maintenance of the equatorial actin filament alignment requires persistent mechanical tension. We therefore quantified equatorial actin filament alignment before and after cortical laser cutting by fluorescence polarization microscopy. We detected a randomly organized actin network at the bottom surface of confined metaphase cells that became increasingly more aligned at later stages of cytokinesis (*Figure 4I*), correlating with an increase in cortical tension along the cell equator (*Figure 4J*). We then laser-cut the actin network perpendicular to the cell equator and measured fluorescence polarization adjacent to the cut region. Importantly, we did not detect significant fluorescence polarization changes induced by laser-cutting at any of the measured cell cycle stages (*Figure 4I*). Thus, maintenance of equatorial actin filament alignment does not require persistent tension.

## Discussion

Prior fluorescence polarization microscopy had shown preferential orientation of actin filaments along the cytokinetic cleavage furrow of vertebrate cells (*Fishkind and Wang, 1993*). However, the absolute degree of filament alignment had not been determined and whether alignment precedes furrow formation, as posited by the canonical anti-parallel filament sliding model, had remained unclear. Our study shows that the initial phase of cleavage furrow ingression is mediated by equatorial contraction of a disordered actomyosin network and that the degree of actin filament alignment within the equatorial cell cortex remains relatively low throughout all stages of furrow ingression, compared to structures like stress fibers or muscle sarcomeres. The late onset and low degree of actin filament alignment within the equatorial cortex suggests a network contraction mechanism underlying cytokinetic cleavage furrow ingression in vertebrate cells, yet our data cannot rule out the possibility that at late cytokinesis stages some actin filaments form antiparallel arrays underneath a randomly oriented actin network.

The quantification of actin filament order at defined time points of cytokinesis has been established through several technical innovations. Previous studies of actin filament order in mitotic cells relied on conventional widefield epifluorescence polarization microscopy, which yields high out-of-focus signal in rounded mitotic cells, restricting accurate quantifications to the flat bottom surface of a cell strain that strongly adheres to the coverslip (*Fishkind and Wang, 1993*). In this prior study, mitotic stages were classified based on visual inspection of the chromosome morphologies, and it has remained unclear how the observed increase in actin filament alignment relates to the ingression state of the cleavage furrow. Our implementation of fluorescence polarization imaging on confocal microscopes strongly reduces out-of-focus signal, enabling to study actin network polarization in the

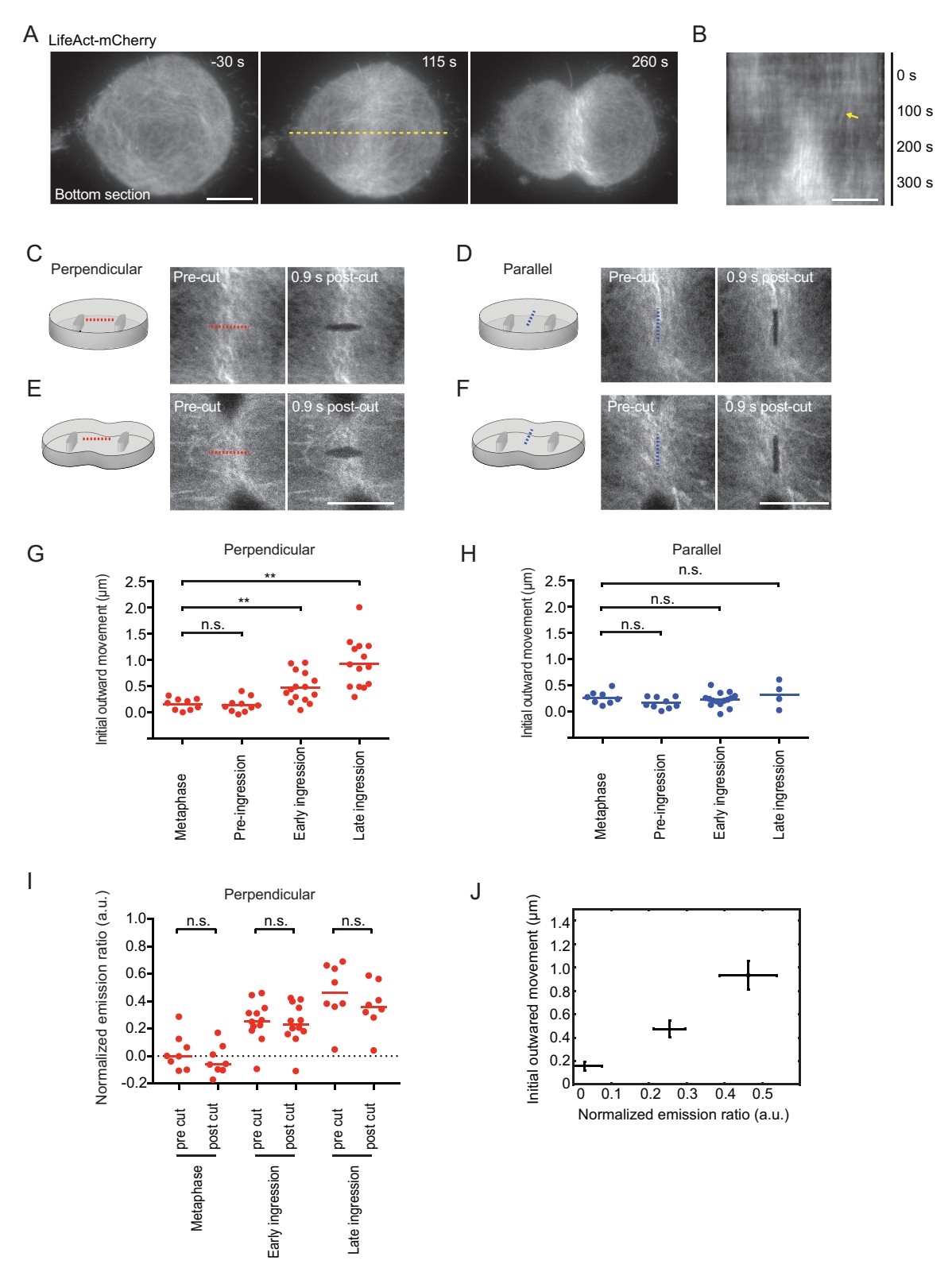

**Figure 4.** Laser microsurgery reveals cortical tension orientation during cytokinesis. (**A**) Total internal reflection fluorescence (TIRF) microscopy of hTERT-RPE-1 under confinement stably expressing LifeAct-mCherry. Representative images of n = 11 cells. Time = 0 s at anaphase onset, as determined in phase contrast images of the same cell. Dashed yellow line indicates position of kymograph shown in (**B**), whereby yellow arrow indicates onset of lateral cortex displacement toward equator. (**C-F**) Laser microsurgery in live hTERT-RPE-1 cells grown in confinement microchambers, at

*Figure 4 continued on next page*

 Research article

*Figure 4 continued*

different stages and orientations as illustrated in the schematics. Cells stably expressing LifeAct-mCherry were cut at their bottom actin cortex along a 5 µm linear path using a pulsed 915 nm laser as indicated by the dashed lines. Red indicates cutting perpendicular to the cell equator circumference, blue indicates parallel cutting orientation. (G) Quantification of cortex tension based on the initial cortical outward movement within 0.9 s after laser cutting perpendicular to the cell equator, for different mitotic stages as indicated. Dots represent individual cells, bars indicate median (metaphase: n = 9, pre-ingression: n = 10, early ingression: n = 15, late ingression: n = 14). (H) Cortical tension quantification for laser cutting parallel to the cell equator (metaphase: n = 8, pre-ingression: n = 9, early ingression: n = 15, late ingression: n = 4). (I) Laser microsurgery as in (C-F), but combined with fluorescence polarization microscopy. Dots represent mean normalized emission ratio values in individual cells measured adjacent to the cutting region before and after laser cutting (metaphase: n = 8, early-ingression: n = 12, late ingression: n = 8; **p<0.0021, n.s. >0.05 by MannWhitney-U test). (J) The increase in initial outward movement in perpendicular cutting experiments (G) correlates with an increase in normalized emission ratio (I). Dots represent mean, bars represent s.e.m. (normalized emission ratio, metaphase = 8, early ingression = 12, late ingression = 8; Initial outward movement, metaphase = 9, early ingression = 15, late ingression = 14). Scale bars = 10 µm.
DOI: https://doi.org/10.7554/eLife.30867.011

The following figure supplement is available for figure 4:

**Figure supplement 1.** Analysis of laser cutting experiments.
DOI: https://doi.org/10.7554/eLife.30867.012

native geometry of rounded mitotic cells. Furthermore, by calibrating the fluorescence anisotropy signal to reference standards with known polarization state, our study informs on the absolute degree of actin filament alignment within the cortical network. A mathematical model for cell geometries established accurate automated quantification of mitotic timing in fixed cells, thereby providing a reference for accurate quantifications of actin network order during cytokinesis progression. Imaging SiR-actin as a live-cell probe for actin filament orientation has further enabled to follow actin network reorganization over time in individual cells. Owing to the relatively slow image acquisition rates on the current confocal polscope implementation, the live-cell set-up can only report on relative changes of actin filament order based on direct ratiometric imaging. Yet, in combination with the fixed cell data, these experiments establish a detailed kinetic profile of actin network organization during cytokinesis progression.

Following cytokinetic cleavage furrow initiation, actin filaments progressively align with the cell equator. This strictly depends on myosin II activity and the degree of actin filament alignment correlates with the local concentration of myosin. This supports a model of actin network reorganization by asymmetric mechanical tension distribution (*Figure 5*). In this model, the activation of myosin II at the cell equator would induce local contractility in a cortical network of randomly oriented actin filaments. Along the cell equator, each given point experiences counteracting contractile forces from neighboring regions, which would build up high levels of mechanical tension along the circumference of the cell. In the direction perpendicular to the cell equator, highly contractile equatorial cortex regions connect to neighboring regions of lower contractility, which would move toward the equator without building high levels of tension. The resulting asymmetric network deformation could explain the filament reorientation toward the direction of the cell equator, which might in turn further increase the contractile forces oriented along this direction.

In *C. elegans* embryos at the one-cell stage, peaks of myosin contractility and cortical actin filament alignment occur at spatially separate locations, whereby long-range cortical flows transduce forces to locally compress the actin filaments before furrowing (*Reymann et al., 2016*). Our data show that in human cells, peaks of myosin contractility and actin filament

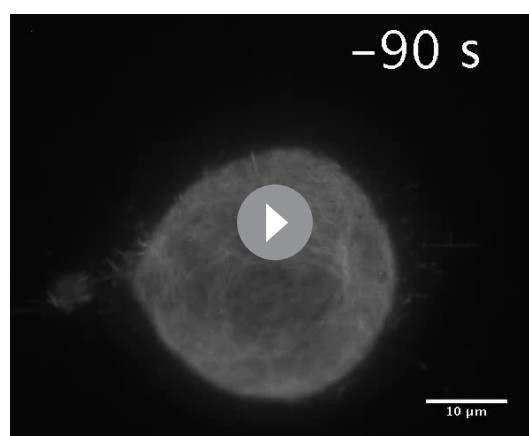

**Video 1.** TIRF microscopy of hTERT-RPE-1 expressing LifeAct-mCherry under confinement stably. Time = 0 s at anaphase onset, as determined in phase contrast images of the same cell. Video shows same cell as *Figure 4A,B*.
DOI: https://doi.org/10.7554/eLife.30867.013

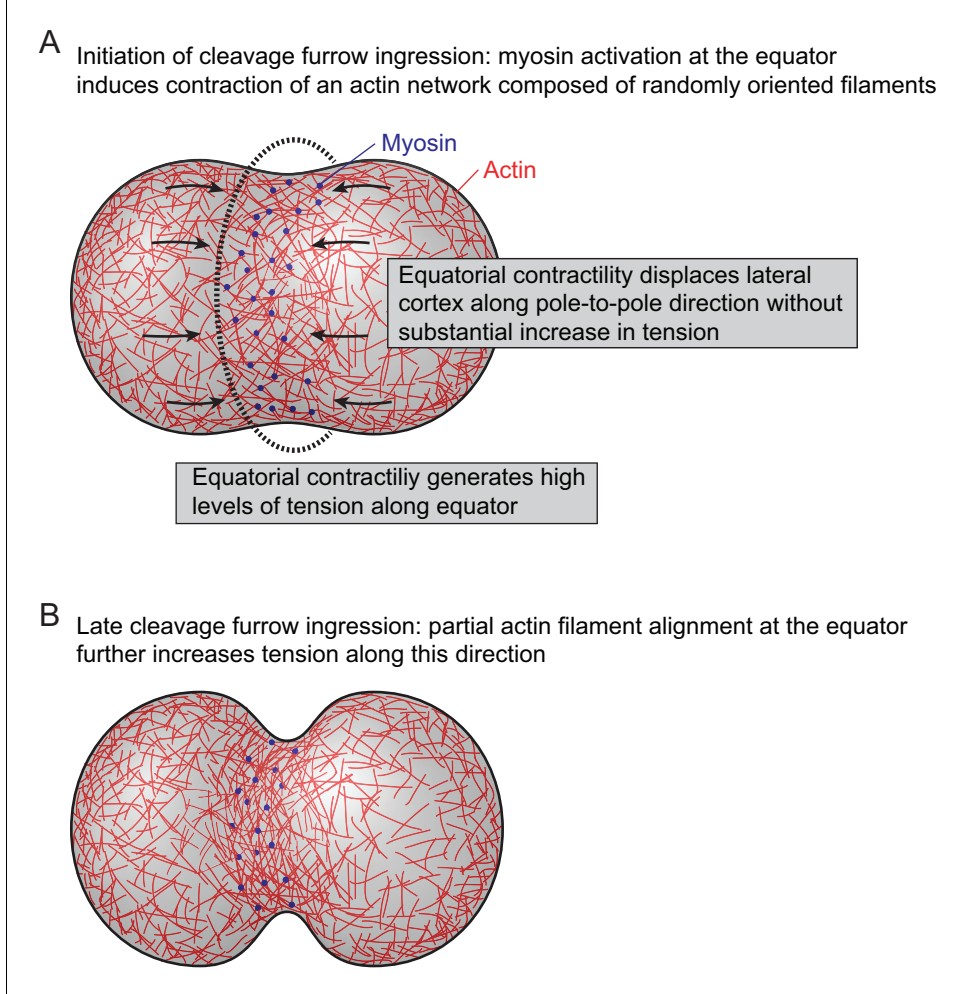

**A** Initiation of cleavage furrow ingression: myosin activation at the equator induces contraction of an actin network composed of randomly oriented filaments

Myosin

Actin

Equatorial contractility displaces lateral cortex along pole-to-pole direction without substantial increase in tension

Equatorial contractiliy generates high levels of tension along equator

**B** Late cleavage furrow ingression: partial actin filament alignment at the equator further increases tension along this direction

**Figure 5.** Model for actomyosin network reorganization during cytokinesis of vertebrate cells. For details see main text.

DOI: https://doi.org/10.7554/eLife.30867.014

alignment co-localize and lateral cortical movement toward the equator does not precede network reorientation. This suggests that during human cell cytokinesis, the actomyosin network aligns directly at the site of highest contractility, whereby lateral cortical movements adjacent to the cell equator (*Cao and Wang, 1990*; *Murthy and Wadsworth, 2005*; *Zhou and Wang, 2008*) might reflect drag imposed by highest network contractility at the equator. In contrast to animal cells, fission yeast cells align actin filaments before contraction of the actomyosin ring (*Stachowiak et al., 2014*; *Wu et al., 2006*; *Vavylonis et al., 2008*; *Laplante et al., 2016*). This might be due to the high mechanical resistance of the cell wall, requiring a higher degree of actin filament reorientation to generate sufficient tension to ingress the plasma membrane.

Fluorescence polarization microscopy previously showed that septin filaments align with the cell equator during late stages of cytokinesis in budding yeast (*DeMay et al., 2011b*; *Vrabioiu and Mitchison, 2006*), but the underlying mechanism has remained unclear. Septins bind actin filaments and are required for proper function of the actomyosin ring in *Drosophila melanogaster* embryos (*Mavrakis et al., 2014*). Based on our observations, it is tempting to speculate that asymmetric distributions of mechanical tension might guide both actin and septin filament systems to self-assemble into polarized networks during cytokinesis.

Mechanical forces have has been suggested to shape other actin-based structures (*Fletcher and Mullins, 2010*). For example, dendritic actin networks respond to external pushing forces by

branching actin preferentially on the convex edge of curved filaments, which orients network growth toward the mechanical force (*Risca et al., 2012*; *Bieling et al., 2016*). While these feedback mechanisms rely on external forces, our data suggest that the actomyosin ring of vertebrate cells responds to intrinsically generated forces. Given that actomyosin rings mediate various other biological processes, including cellular wound healing and developmental morphogenesis (*Schwayer et al., 2016*), it will be interesting to further investigate the intricate relationship between contractile forces and actin network organization.

## Materials and methods

### Cell lines and cell culture

All cells used in this study were derived from immortalized human retinal pigment epithelial cells (hTERT-RPE-1, ATCC, CRL-4000). This cell line was authenticated by a Multiplex human Cell line Authentication test (MCA). All cell lines used in this study have been regularly tested for mycoplasm contamination with negative results. The following monoclonal hTERT-RPE-1 cell lines expressing fluorescent proteins were generated: H2B-mRFP-hTERT-RPE1, LifeAct-mCherry-hTERT-RPE1, MRL2B-EGFP-hTERT-RPE1, Actin-GFP-hTERT-RPE1, MyrPalm-mEGFP-hTERT-RPE1. Cell lines were cultured in Dulbecco's modified Eagle medium (DMEM; Gibco, Carlsbad, CA) supplemented with 10% (v/v) fetal bovine serum (FBS; Gibco), 1% (v/v) penicillin-streptomycin (Sigma-Aldrich, St. Louis, MO, USA), 500 µg ml$^{-1}$ G418 (Gibco) and 0,5 µg ml$^{-1}$ puromycin (Calbiochem, San Diego, CA). For live-cell imaging cells were grown on LabTek II (Thermo Fischer Scientific, Boston, MA) chambered coverglass or on 50-mm glass-bottom dishes cell culture dishes (MatTek Corporation, Ashland, MA). Live-cell imaging was carried out in DMEM containing 10% (v/v) FBS and 1% (v/v) penicillin/streptomycin, without phenol red and riboflavin to reduce autofluorescence.

### Cell confinement microchambers

Cell confinement microchamber was fabricated as described in *Le Berre et al. (2014)*. Briefly, hard micropillars made of PolyDiMethylSiloxane (PDMS, Sylgard 184 Silicone Elastomer Kit, Dow Corning, Midland, MI) with a ratio of 8:1 of elastomer:curing agent were disposed onto a 5-µm-height micropillar mold produced by photolithography (IST Austria, Jack Merrin). Subsequently plasma-cleaned coverslips (Menzel-Gläser, VWR, Radnor, PA) were placed onto the mold, pressed, and backed onto a 95°C heating plate for 15 min to generate micropillar-coated coverslips. Cell confinement was performed in 50-mm glass-bottom dishes (MatTek Corporation, Ashland, MA), which were plasma cleaned for 90 s and coated with 20 µg/ml fibronectin diluted in PBS overnight prior to seeding cells. Soft PDMS pillars (30:1 elastomer:curing agent) were attached to micropillar coated coverslips, inverted and mounted on top of the 50-mm glass-bottom cell culture dish. A home-made magnetic lid was used to apply homogeneous pressure on the micropillar coated coverslip to ensure homogeneous and reproducible cell confinement.

### Plasmid transfection

Plasmids were transfected into hTERT-RPE-1 cells using X-tremeGENE9 DNA transfection reagent (Roche, Basel, Switzerland) according to the manufacturer's instructions. Imaging was performed 48 hr post-transfection.

### Inhibitors and stains

Para-nitroblebbistatin (Optopharma, Budapest, Hungary) was added 15 min prior to cell imaging to a final concentration of 50 µM (*Figure 2B,G*, *Figure 3—figure supplement 1A,C*). SiR-actin (Spirochrome, Stein am Rhein, Switzerland) was used at a final concentration of 0.1 µM (*Figure 2A,B,D–G*, *Figure 2—figure supplement 1C–F,H*, *Figure 2—figure supplement 2*, *Figure 3—figure supplement 1*, *Figure 2—figure supplement 3*) or 0.2 µM (*Figure 1—figure supplement 1H–J*), 0.3 µM (*Figure 3E–G*, *Figure 4I,J*). Hoechst 33342 (Sigma-Aldrich, St. Louis, MO) was used at a final concentration of 0.2 µg ml$^{-1}$ (*Figure 4A–F*, *Figure 4—figure supplement 1A,G*).

## Indirect immunofluorescence staining of cells synchronized to cytokinesis

Wild-type hTERT-RPE-1 cells were grown in T75 flasks (Thermo Fischer Scientific, Boston, MA) to 80% confluency. Then, dead cells were removed by shake-off and replacement of the culture medium. Mitotic cells were collected 2 hr later by shake-off and seeded on coverslips (Carl Roth, Karlsruhe, Germany) coated with 0.01% poly-L-lysine (Sigma-Aldrich, St. Louis, MO) in DMEM medium. 15 min after mitotic shake-off, cells were fixed by 4% (w/v) formaldehyde solution (methanol-free, Thermo Fischer Scientific, Boston, MA) in cytoskeleton buffer (20 mM Pipes, 2 mM $MgCl_2$, 10 mM EGTA pH 6.8). Cells were permeabilized by 0.1% Triton X-100 in cytoskeleton buffer for 2 min and subsequently blocked for 30 min using BlockAid (Thermo Fischer Scientific). Actin was stained with AlexaFluor 488 phalloidin (Thermo Fischer Scientific) at a final concentration of 0.25 µM for 30 min in BlockAid and DNA was labeled with DAPI (Thermo Fischer Scientific) at a final concentration of 0.1 µg ml$^{-1}$ for 10 min. Stained coverslips were washed with PBS and mounted using Vectashield antifade mounting medium (Vector Laboratories, Burlingame, CA).

## Microscopy

### Confocal microscopy

Confocal laser scanning microscopy was performed on a Zeiss LSM780 microscope using a 40 × 1.4 NA Oil DIC Plan-Apochromat objective (Carl Zeiss, Jena, Germany) or a Zeiss LSM710 microscope, using 63 × 1.4 NA Oil DIC Plan-Apochromat objective (Carl Zeiss), both controlled by ZEN 2011 software. Fast time-lapse imaging was performed on a spinning-disk confocal microscope (UltraView VoX, Pelkin Elmer, Waltham, MA) with a 100 × 1.45 NA alpha Plan-Fluar objective (Carl Zeiss), controlled by Volocity 6.3 software (Perkin Elmer, Waltham, MA). All three microscopes were equipped with an incubation chamber (European Molecular Biology Laboratory (EMBL), Heidelberg, Germany), providing a humidified atmosphere at 37°C with 5% $CO_2$.

### Confocal fluorescence anisotropy microscope

Fluorescence polarization microscopy in live cells was performed on a Zeiss LSM780 using a x40, 1.4 NA. Oil DIC Plan-Apochromat objective (Carl Zeiss, Jena, Germany) or on a Zeiss LSM710 using 63 × 1.4 N.A Oil DIC Plan-Apochromat objective (Carl Zeiss). Both systems were controlled by ZEN 2011 software and were equipped with an incubation chamber (EMBL, Heidelberg, Germany), providing a humidified atmosphere at 37°C with 5% $CO_2$. For fluorescence anisotropy measurements, specimen was excited with 90 degree polarized laser light (relative to the X-axis of the optical table) two images were sequentially recorded with linear polarizers in the emission light path oriented 0° or 90° relative to the X-axis of the optical table, respectively.

### Confocal liquid crystal-based fluorescence polarization microscope (LC-PolScope)

Confocal fluorescence polarization microscopy of fixed cells was performed on a Zeiss LSM780 microscope equipped with a LC universal compensator (OpenPolScope, Woods Hole, MA) used as a variable linear polarizer, introduced into the illumination path using the analyzer slider slot. The LC universal compensator was controlled by the OpenPolscope software package (Micro-Manager 1.4.15, OpenPolScope 2.0). The LC universal compensator was synchronized to image acquisition with ZEN 2011 using the OpenPolScope software package. 3-D-image stacks were recorded with 5–15 z-sections.

### Laser microsurgery

Laser microsurgery was performed on a Zeiss LSM710 confocal microscope equipped with two linear polarization filters in the emission light path and a tuneable Chameleon Ti:Sapphire laser for multi-photon excitation using a 63 × 1.4 N.A Oil DIC Plan-Apochromat objective (Carl Zeiss, Jena, Germany), controlled by a ZEN 2011 software. The microscope was equipped with an incubation chamber (EMBL, Heidelberg, Germany) providing a humidified atmosphere at 37°C with 5% $CO_2$. The cell cortex was cut using a tuneable Ti:Sapphire laser set to 915 nm and 100% power with 25 µs dwell time along a 5 µm linear path. To achieve reproducible cutting conditions, the laser was always

moved along a horizontal path, using only cells that had a spindle axis of 90° ± 20° and 0° ± 20° relative to the X-axis of the optical table.

## Fluorescence recovery after photobleaching (FRAP)
Fluorescence after photobleaching was performed on a Zeiss LSM 780 microscope using 63 × 1.4 NA Oil DIC Plan-Apochromat objective (Carl Zeiss, Jena, Germany). hTERT-RPE1 cells stably expressing Actin-GFP were grown in 5 µm confinement chambers and fluorescent signal was photobleached by moving a 488 nm laser on a 5 µm linear path with 50 µs dwell time per pixel along the cell cortex. Fluorescence recovery was monitored by acquiring time lapse movies after photobleaching.

## Total internal reflection fluorescence microscopy
Total internal reflection fluorescence microscopy (TIRF) was performed on a customized iMIC stand (TILL Photonics, FEI, Hillsboro, OR) equipped with a multipoint 360° TIRF and an Olympus 100 × 1.49 NA high-performance TIRF objective. For LifeAct-mCherry excitation, an acousto-optical tunable filter (AOTF)-modulated 561 diode-pumped solid-state lasers laser (DPSS) line was used. The incidence angle was adjusted using a two-axis scan head that was rotated around 360°. Images were collected with an Andor iXON DU-897 EMCCD camera (Andor Technology Ltd. Belfast, UK) controlled by the Live Acquisition software (TILL Photonics, FEI, Hillsboro, OR). The microscope was equipped with a homemade heating stage and objective heating collar (IST-A machine and e-machine shop) to incubate cells at 37°C and with a humidified atmosphere with 5% $CO_2$ (ibidi GmbH, Martinsried, Germany).

## Image analysis
### Staging of fixed cells relative to anaphase onset by cell geometry-based model
To estimate the time that cells were chemically fixed relative to anaphase onset, a regression model based on chromatin-chromatin distance and cleavage furrow diameter was developed. hTERT-RPE-1 cells stably expressing H2B-mRFP were stained with SiR-actin and imaged live every 15 s from metaphase until full ingression of the cytokinetic cleavage furrow. In these cells, the distance between the centers of the two segregating chromatin masses and the cleavage furrow diameter was measured at each time point (*Figure 1—figure supplement 2A,B*). A polynomial function with degree d = 4 was fitted to cleavage furrow diameter and chromatin-chromatin distance. The fitted regression model was used to infer the time after anaphase onset for experiments with fixed cells. Cells were classified to four stages (*Figure 1E*): metaphase, pre-ingression (<100 s), early ingression (101–150 s), mid ingression (151–200 s) and late ingression (201–301 s) relative to anaphase onset.

### Quantification of fluorophore alignment with confocal LC-PolScope
Images acquired with the confocal LC-PolScope were analyzed using Fiji (version 1.47 s) in combination with a semi-automated pipeline developed in Matlab (MATLAB Release 2013b, The MathWorks, Inc, Natick, MA). Fluorescence anisotropy was calculated for image stacks of four frames that were sequentially recorded with linearly polarized light oriented at 0°, 45°, 90°, and 135°. A fifth image with 0 degree was recorded at the end to enable correction for acquisition photobleaching, as described in *DeMay et al. (2011a)*. To obtain the 'anisotropy image', we calculated the polarization factor ($p$) for each camera pixel based on the intensities in the images measured at four different linear polarization states as following:

Polarization at the polar and equatorial cell cortex was determined using a custom developed Matlab analysis pipeline (Supplemental *source code 1*) as follows: we segmented the cell by a seeded watershed algorithm using the average image of all four different angle images as input. We next determined the equatorial and polar center positions by fitting an ellipse to the segmented contour and using the intersection between the segmented contour and long/short axis of the ellipse as the equatorial and polar regions. All identified regions were visually inspected and corrected if required. Using the intersecting pixel as the center we divided the segmented cortex outline with a width of 640 nm into 2 µm regions at the equator and 16 µm regions at the poles. This region was then transferred to the anisotropy image and intensities within this region were averaged

(*Figure 1—figure supplement 1B*). These values were subsequently normalized between reference values separately measured for fluorescent plastic and AlexaFluor 488-phalloidin-stained stress fibers (*Figure 1C,G*) to calculate the normalized polarization factor $p_{norm}$:

$$a = (I_0 - I_{90}), \; b = (I_{45} - I_{135}), \; c = (I_0 + I_{45} + I_{90} + I_{135})$$

$$p = \frac{\sqrt{a^2 + b^2}}{0.5c}$$

$$p_{norm} = \frac{p - p_{plastic}}{p_{stressfibers} - p_{plastic}}$$

The width of the anisotropic zone at the cell equator (*Figure 1F*) was determined by fitting a Gaussian function to the polarization factor line profiles in late-stage furrow ingression cells.

## Quantification of fluorescence anisotropy in images recorded with linear polarizers in emission light path

Fluorescence polarization microscopy of live mitotic cells was performed using two linear polarizers in the emission light path of the confocal microscope. The use of a polarized laser excitation beam to excite fluorescent dipoles is incompatible with analysis using standard procedures as previously reported (*Fishkind and Wang, 1993*). We therefore measured relative changes of actin filament orientation over time based on a simple ratio between the images acquired with the horizontal and the vertical emission polarizers. The analysis pipeline for calculating the fluorescence emission ratio between the two images was then adapted to the respective cellular context (bottom section of a confined cell or central section of an unconfined cell) as detailed below.

In general, images were then analyzed using Fiji and a semi-automated analysis pipeline in Matlab. Global differences in transmission through the two linear polarizers were determined by imaging fluorescent plastic slides using both linear emission polarizers (isotropic fluorophore dipole sample that should be identical under both conditions). The channel transmission factor (CTF) was calculated by dividing the image acquired by using the horizontal linear polarizer by the image acquired with the vertical linear polarizer. This correction factor was multiplied with the image acquired with the vertical linear emission polarizer prior to downstream analysis.

For quantification of emission ratio at the bottom surface of cells (*Figure 3E*, *Figure 4I*), the mean fluorescence intensity in a region outside the cell was measured and subtracted from each frame of the respective image prior to analysis (background correction). We then calculated the ratio between images acquired with 0° and 90° linear emission polarizers in a rectangle region (1 μm x 4 μm) centered at the cell equator (*Figure 3E*). For emission ratio measurements after laser cutting, we calculated the ratio between images acquired with 0° and 90° linear emission polarizers in squared regions (2 μm x 2 μm) perpendicular and adjacent to each cut border. The emission ratio obtained from the four regions two on each side of the cut was averaged (*Figure 4I*) and subsequently normalized to a reference scale between emission ratio values derived from metaphase cortices (representing a random actin network, see *Figure 1C*), and stress fibers (representing aligned actin filaments). Stress fiber values were measured by manually drawing a 120 pixel long and 5 pixel wide line along a stress fiber using Fiji. Final values were derived by calculating the median from all measured values.

Unlike for emission ratios derived from planar surfaces at the bottom section, we must consider the cell cortex orientation for emission ratios measured at central z-sections. This is because the emission ratio is determined not only by the fluorescent dipole orientation relative to the linear polarizers, but also by the fluorescent dipole orientation relative to the polarization of the exciting laser light. Only cells with a spindle axis of 90° ± 20° (relative to the X-axis of the optical table) were used for the measurements. To correct for the contribution of emission ratio changes caused by the polarized excitation light, we recorded reference images of metaphase cells stained with SiR-actin and measured the fluorescence ratio along the cortex (*Figure 2—figure supplement 1D*). Because the actin network of metaphase cortices is randomly oriented, changes in emission ratio are only caused by the fluorescence dipole orientation relative to the excitation beam. We developed a regression model for excitation light dependent emission ratio changes by fitting a sine squared

function with a wavelength to 180° to the emission ratio along the metaphase cortex (*Figure 2—figure supplement 1E*). To correct for excitation light contribution, we divided the observed ratio by the regression model and obtained a ratio of 1 along the entire metaphase cortex (*Figure 2—figure supplement 1F*). We then applied this regression model to the dividing cells by integrating it into the following workflow: we first segmented the cell cortex by using a seeded watershed algorithm (position 0 µm marks the furrow midpoint in all subsequent plots). We then divided the segmented cortex in four sub-regions of 300 pixels length using the furrow or pole position as the center of the segment (*Figure 2—figure supplement 2A*). The cortex orientations at individual pixel positions was inferred by fitting a B-spline to the segmented outline of each cell for every time frame (*Figure 2—figure supplement 2B*). This segmentation contour line had a width that completely contained the cortex signal even when the two image channels were slightly shifted owing to cleave furrow ingression during their sequential acquisition (e.g. *Figure 2B*, control, 330 s). We next averaged the fluorescence intensity over seven pixels perpendicular to the tangential vector using the segmented pixel as the central pixel (*Figure 2—figure supplement 2B*). Averaged intensity values for each central pixel were used to calculate the ratio between linear emission polarizers of 0° and 90° to obtain the observed ratio (*Figure 2—figure supplement 2B*). By evaluating the regression model for the pixel orientation along the segmented contour, we obtained the expected ratio that is purely based on the cortex geometry (see also *Figure 2—figure supplement 1D–F*). We next divided the 'observed ratio' for each averaged pixel of the contour by the 'expected ratio' to correct for cell geometry effects (*Figure 2—figure supplement 2B*) and calculate a 'corrected ratio'. Regions far away from the furrow midpoint at 0 µm in this 'corrected ratio' plot show emission ratio values close to one as expected for a randomly organized actin filament network, while emission values at the cleavage furrow show lower values indicating actin filament alignment (*Figure 2—figure supplement 2B*). To reduce noise, the emission ratio was averaged within a 1 µm window centered at the cell equator and at the cell poles. The final 'normalized' emission ratio was calculated by dividing the 'corrected emission ratio' by the average of the emission ratio of three metaphase frames to obtain changes relative to anaphase onset.

For display of normalized emission ratio values on straight contours (*Figure 1—figure supplement 1C*), we used a custom-developed Matlab script to extract pixel intensity values along the segmentation mask. Pixel intensity values were then normalized between 1 and 0 by first subtracting the lowest pixel value from the image and then by dividing the image by the brightest pixel value. The Fiji plugin 'straighten' was then used to generate linescan images.

The following section further details the calculations described above:

Outline of a mitotic cell was segmented using a seeded watershed algorithm (*Equation 1*)

$$O_{Mitotic} \longleftarrow Segmented\ outline\ mitotic\ cell \tag{1}$$

B-spline (BS) fit was used to determine the tangential angle (α) of each pixel of the cortical outline (*Equation 2*)

$$\alpha(x) = BS(x),\ x \in O_{Mitotic} \tag{2}$$

Pixel intensities acquired with horizontal linear emission polarizer were divided by pixel intensities acquired with vertical linear emission polarizer, leading to a measure of fluorescence ratio (*r*). To avoid bias in the fluorescence ratio, the transmission differences between the two channels needs to be corrected. We therefore measured channel transmission factor (*CTF*) using a specimen with randomly oriented fluorescent dipoles. Because of the polarized nature of both the excitation light and the emission light, the resulting fluorescence ratio is influenced by the dipole orientation relative to angle of the excitation light. To correct for this, we measured the fluorescence ratio along the cortex of a metaphase cells.

$$r_{Mitotic}(x, \alpha) = \frac{I_{\perp Mitotic}(x, \alpha)}{I_{\| Mitotic}(x, \alpha) \cdot CTF} \tag{3}$$

Similar quantities were defined for metaphase cells.

$$O_{Metaphase} \longleftarrow Segmented\ outline\ metaphase\ frames \tag{4}$$

$$\alpha(x) = BS(x), \ x \in O_{Metaphase} \tag{5}$$

$$r_{Metaphase}(x, \alpha) = \frac{I_{\perp Metaphase}(x, \alpha)}{I_{\parallel Metaphase}(x, \alpha) \cdot CTF} \tag{6}$$

Next, we built a regression model of angle-dependent emission ratio (*R*) based on segmented metaphase cortices

$$R_{Metaphase}(\alpha) = a_0 + a_1 \cdot \sin^2\left(r_{Metaphase}(x, \alpha) \cdot \left(\frac{\pi}{180}\right)\right) \tag{7}$$

Above experimentally determined regression was then used to correct the angle dependence in data.

$$r_{corrected}(x, \alpha) = \frac{r_{Mitotic/Metaphase}(x, \alpha)}{R_{Metaphase}(\alpha)} \tag{8}$$

Finally, the corrected normalized emission ratio values were divided by pre-anaphase values.

$$r_{Norm} = \frac{r_{corrected}}{r_{pre-anaphse}} \tag{9}$$

## Cell geometry quantification

Cleavage furrow diameter and pole-pole distance was measured manually in the central section of a Z-stack. In cell confiner experiments, the cleavage furrow diameter along the optical axis was measured by sectioning the Z-stack in the axial dimension along a line centered at the midpoint of a cell. The distance between top and bottom surfaces was measured manually at the cell equator.

The distance between the segregating chromosome masses during anaphase was measured based on the H2B-mRFP label. The H2B-mRFP images were segmented using a custom-developed MATLAB script and the weighted center of mass was calculated for each segregating mass of chromosomes. Prior to sister chromatid separation, only a single chromatin center position was determined.

Cell cortex curvature was calculated by fitting a circle to the center position of the cell equator of the segmented outline of a cell. Curvature was calculated as the reciprocal of the radius of the circle. When the midpoint of the circle was located inside the cell the curvature was defined as negative, and if the midpoint of the circle was located outside the cell it was positive.

## Quantification of actin and myosin fluorescence at the cell cortex

SiR-actin and myosin regulatory light chain 12-B-EGFP (gift from Roland Wedlich-Soeldner, University of Muenster, Germany) fluorescence intensities were measured on cortices segmented using a seeded watershed algorithm. Line profiles along the entire cell cortex were generated based on the segmented cortex contour. The average of cytoplasmic and extracellular fluorescence was subtracted from each image prior to analysis of cortical fluorescence. Cortical fluorescence intensities were averaged within a 4 µm region centered at the cell equator or at the cell poles, respectively. These values were normalized to anaphase onset (the last time frame in which chromosomes were not separated) by dividing by the average value determined in three time frames prior to anaphase onset. For each individual cell, an average line profile was calculated based on time points from 180 s after anaphase onset until full ingression, normalized to the maximum value. Line profiles from 14 cells were subsequently averaged (*Figure 2—figure supplement 3D,E*).

## Cortex thickness measurement

RPE1 cells expressing MyrPalm were stained for 2 hr with 200 nm SiR-actin. Interphase cells were trypsinized for 1 min and quenched with DMEM immediately before imaging to obtain rounded cells while metaphase cells and late anaphase cells were obtained from asynchronously growing cells. Two-colour images (green and far-red) were recorded on an LSM780 using a 63 x (1.4 NA (numerical aperture)) objective. For chromatic correction 500 nm diameter multicolor TetraSpeck fluorescent beads (Invitrogen T14792) were imaged using the same settings. After correction for the chromatic

shift in x and y (73.5 nm and 39.8 nm, respectively) linescans were performed in Fiji. For this, we measured a 10 px wide line perpendicular to the cell cortex in a region devoid of filopodia. For measurements in anaphase cells, only cells with substantial furrow ingression were used and the line was positioned in the furrow (see *Figure 2—figure supplement 1G*). After background substraction a Gaussian function was fitted to the linescans using a custom developed Matlab script. The fit area was restricted to a 11 pixel wide region, using the central pixel as the maximum value of the linescan. The distance between the Gaussian center positions for MyrPalm and Sir-Actin fluorescence channels determined ($X_{MyrPalm} - X_{Sir-actin}$). The cortex thickness is obtained by multiplying this value by two (see *Clark et al., 2013*).

## Kymograph measurement
Kymographs were generated in Fiji, by using the KymoResliceWide plugin after drawing a straight 25 px wide linear line from cell pole to cell pole.

## Quantification of cortical tension in laser microsurgery experiments
The initial outward movement of the cell cortex after laser cutting was determined by measuring the distance between the edges adjacent to the cut in the first frame after cutting. Outward movement is composed mainly of myosin-induced contractility (*Fischer-Friedrich et al., 2016*), but also of laser-induced damage. To estimate the contribution of laser induced damage to the initial outward movement, we measured the position of landmark structures before and after cutting in cells in which non-muscle myosin-II activity was inhibited by addition of 50 µM para-nitroblebbistatin. The average value for landmark displacement pairs was used as a baseline that was subtracted from the measured distance between the edge of displaced cortex after cutting. hTERT-RPE-1 cells were classified into four cell division stages based on the ratio of the cleavage furrow diameter and the cell diameter at the position of the chromosome masses: metaphase (pre-chromosome segregation), pre-ingression (ratio >1), early ingression (ratio <1 and>0.8), late ingression (ratio <0.8).

## Acknowledgements
The authors thank Hervé Turlier for comments on the manuscript, the IMBA/IMP/GMI BioOptics core facility for technical support, and Life Science Editors for editorial support. DWG has received funding from the European Community's Seventh Framework Programme FP7/2007-2013 under grant agreement no. 241548 (MitoSys) and no. 258068 (Systems Microscopy), an ERC Starting Grant under agreement no. 281198 (DIVIMAGE), and from the Austrian Science Fund (FWF) project no. SFB F34-06 (Chromosome Dynamics). FS has received funding from an EMBO long-term fellowship (ALTF 1447–2012). SM has received funding from Human Frontier Science Program cross-disciplinary fellowship (LT000096/2011).

# Additional information

### Funding

| Funder | Grant reference number | Author |
| --- | --- | --- |
| European Commission | 241548 | Daniel W Gerlich |
| Austrian Science Fund | SFB F34-06 | Daniel W Gerlich |
| European Research Council | 281198 | Daniel W Gerlich |
| European Commission | 258068 | Daniel W Gerlich |
| European Molecular Biology Organization | ALTF 1447-2012 | Felix Spira |
| Human Frontier Science Program | LT000096/2011 | Shalin Mehta |

The funders had no role in study design, data collection and interpretation, or the decision to submit the work for publication.

## Author contributions
Felix Spira, Conceptualization, Software, Formal analysis, Validation, Investigation, Visualization, Methodology, Writing—review and editing, Conception and design, acquisition of data, analysis and interpretation of data, drafting or revising the article; Sara Cuylen-Haering, Formal analysis, Validation, Investigation, Visualization, Methodology, Acquisition of data and analysis and interpretation of data; Shalin Mehta, Software, Formal analysis, Investigation, Methodology, Writing—review and editing, Acquisition of data, analysis and interpretation of data, drafting or revising the article; Matthias Samwer, Formal analysis, Investigation, Visualization, Writing—review and editing, Acquisition of data, drafting or revising the article; Anne Reversat, Resources, Methodology, Acquisition of data, contributed unpublished essential data, analytical tools, or reagents; Amitabh Verma, Resources, Contributed unpublished essential data, analytical tools, or reagents; Rudolf Oldenbourg, Michael Sixt, Resources, Supervision, Contributed unpublished essential data, analytical tools, or reagents; Daniel W Gerlich, Conceptualization, Formal analysis, Supervision, Funding acquisition, Validation, Methodology, Writing—original draft, Project administration, Writing—review and editing, Conception and design, analysis and interpretation of data, drafting or revising the article

## Author ORCIDs
Felix Spira http://orcid.org/0000-0001-5490-5508
Sara Cuylen-Haering http://orcid.org/0000-0002-1193-4648
Daniel W Gerlich http://orcid.org/0000-0003-1637-3365

## Decision letter and Author response
Decision letter https://doi.org/10.7554/eLife.30867.019
Author response https://doi.org/10.7554/eLife.30867.020

# Additional files

## Supplementary files
• Source code 1.
DOI: https://doi.org/10.7554/eLife.30867.015

• Transparent reporting form
DOI: https://doi.org/10.7554/eLife.30867.016

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
