## [Decision Letter]

[Editors’ note: a previous version of this study was rejected after peer review, but the authors submitted for reconsideration. The first decision letter after peer review is shown below.]

Thank you for submitting your work entitled "Actomyosin contracts as a network to drive cytokinesis in vertebrate cells" for consideration by *eLife*. Your article has been reviewed by three peer reviewers, one of whom is a member of our Board of Reviewing Editors and the evaluation has been overseen by a Senior Editor. The reviewers have opted to remain anonymous.

Our decision has been reached after consultation between the reviewers. Based on these discussions and the individual reviews below, we regret to inform you that your work will not be considered further for publication in *eLife*.

The referees were uniformly positive about the field and the fact that you are using cutting edge tools to investigate the idea originally proposed over 20 years ago by Fishkind and Wang. One of the referees felt that the study would benefit from cutting experiments after low dose Blebbistatin treatment. However, while other reviewers agreed this was a good experiment, they raised too many concerns with how the experiments were done and would like to have seen the raw data as well as experimental details described better. An extract of the discussion is also included for your information. In light of these concerns, we are unable to proceed further. We will be willing to consider a new submission with all details provided, which will be seen by the three referees. I will point out that there will be no guarantees of publication, since the conclusions may or may not hold.

Laser cutting after drug treatment is itself an interesting experiment. However, I feel that this can't correct the major defects of this manuscript such as:

– Lack of raw data: no raw images of the polarity microscopy of a dividing cell have been shown.

– Questionable methodologies for both data acquisition and analysis (inconsistent definitions of anisotropy etc.)

– Oversimplified assumption that all the actin filaments are parallel to the membrane and there is no thickness of the cortex.

Under this circumstance, it is impossible to suggest a defined set of revisions that would sort of guarantee the publication.

Reviewer #1:

Spira et al., present a valuable new approach to spatiotemporally monitor changes in actin filament organization during cleavage furrow ingression. Previously, EM and quantitative fluorescence microscopy–based assays have visualized the formation of discreet foci of actin filaments prior to cleavage furrow formation. The method proposed here, fluorescence polarization microscopy, provides an opportunity to monitor dynamics of actin filament organization during cleavage furrow ingression and link it directly to ring contraction in a live setting. Based on their work the authors conclude that the, (a) actin reorientation appears not to be critical for transducing forces leading to the early stages of cleavage furrow ingression, (b) actin filament reorientation is a consequence of mechanical input from myosin II, and (c) filament reorientation is a consequence of the mechanical coupling along the contractile ring. However I have a few concerns that I would like the authors to address before publication.

The results presented in this manuscript are largely consistent with those observed in other cellular systems such as sea urchin and fission yeast, namely that filament reorganization occurs much later during cleavage furrow ingression and is not necessarily a cause for the initiation of furrow formation which reduces the novelty of the authors' conclusions.

That the actin network transforms from a largely isotropic cortical meshwork in metaphase cells into an anisotropic network shortly after the onset of anaphase onset has already been reported earlier (see Fishkind and Wang, 1993). Understandably, the present work provides important dynamic insights along with data to suggest presence of radial forces along the cell equator that drive ring contraction and as such is interesting.

However the authors should clarify the following concerns before publication. In general, the description of their methods needs better articulation. In particular I would like the authors to clarify on the following points:

Definition of polarization factor p is not clear. Is it the same as anisotropy?

What is r_norm in formula (9)? Is it normalization polarization factor? It is very hard to follow since the description of methods is poorly written.

According to the definition of anisotropy (subsection “Quantification of fluorophore alignment with Confocal LC-PolScope”), why would a random network have 50% anisotropy (Figure 1)? I would expect I_0 = 1, I_90 = 0, I_45 = 0.5, I_135 = 0.5 and hence a = 1, b = 0, c = 2 and therefore anisotropy = 1.

Figure 1: Early furrow ingression, the polarization factor varies in a huge range. It is therefore unreliable to say the network is still random.

Figure 1: looks like anisotropy starts to reduce at 60 sec while ingression starts at ~ 90 sec (Figure 1). So it does not look like "Contraction of an isotropic actin network initiates cleavage furrow ingression" as claimed.

Discussion section, "Our study shows that disordered actomyosin networks can generate sufficient mechanical tension to initiate cleavage furrow ingression in vertebrate cells": This is of low confidence as mentioned above.

Subsection “Equatorial actin filaments reorient on planar cortex regions”: "While these data do not rule out that local furrowing facilitates actin filament alignment, they show that planar network contraction is sufficient to induce substantial alignment." Comment: the network is not only simply planar, but also part of the ring. Alignment would occur along the ring (and this is expected), including the part on the planar network.

Figure 4, post–cut time should be reported to tell whether cells were imaged at the same post–cut time.

"Mechanical coupling" is mentioned at a few places, but what does it mean? What is coupled to what?

Title: "Actomyosin contracts as a network to drive cytokinesis in vertebrate cells". Comment: The data did not convince if the whole network contracts or part of it forms a ring which then contracts.

Reviewer #2:

Animal cells execute cytokinesis by ingression of the cleavage furrow that is driven by contraction of the ring–like actomyosin network at the cell equator. The details of the dynamic organization of actin filaments in the ring are not well understood. The authors employed fluorescence polarization microscopy to determine the anisotropy of actin filaments in fixed and live cells, and detected gradual alignment of actin filaments during furrow deepening. They also performed laser microsurgery experiments and discovered an anisotropy in the cortical mechanical tension at the cell equator.

Although Fishkind and Wang, 1993 already demonstrated an increase of anisotropy of actin filaments at the equatorial cortex during cytokinesis of mammalian cultured cells using fluorescence polarization microscopy in the 90's, it was in fixed cells with a non–confocal set–up and thus the z– and time–resolution was not high. It is of significant interest to revisit this issue using modern microscopy technology, which has progressed in the last two decades. Indeed, the images of fixed and live interphase cells by a confocal LC–PolScope (Figure 1) are amazing. However, unfortunately, there are many points to be clarified or improved as below, which would prevent publication of this interesting work in the current form. The experimental evidence doesn't seem to be sufficient for discussions with physical terms such as "dissipation" or" force feed–back".

1) Fluorescence polarization microscopy is more powerful for the anisotropy in the X–Y plane than that in a plane parallel to the Z–plane. Indeed, Fishkind and Wang (1993) examined the anisotropy in the X–Y plane in a similar scheme to that used for Figure 3 and 4. Why wasn't the confocal LC–PolScope used for the observation of the bottom of the cell (after fixation if necessary)? How significant is the motion artifacts in dividing cells?

2) In general, the anisotropy measurement depends on the relative scale between the spatial frequency of the target structure and the spatial resolution of the imaging. Imagine an infinitely repeating orthogonal grid of 100 µm interval. If the measurement is done at the scale of 1 mm or larger, the anisotropy of this structure would be uniformly 0. On the other hand, at the resolution of 10 µm, the anisotropy would be non–uniform, i.e., >0 on a line while 0 between the lines. This would result in >0 value after spatial averaging. This means that, even for the same structure, a smaller anisotropy value would be obtained when the spatial resolution becomes lower. Conversely, with a constant spatial resolution, isotropic shrinkage of a network structure would result in a smaller anisotropy value.

In the actual measurement of the cortical actin network, the distribution of the spatial frequencies is complex and can range from ~7 nm (the diameter of a single actin filament), 10~100 nm (filament–to–filament distance in a bundle such as a microvillus, a sarcomere or a stress fiber), or bigger (~µm). The imaging resolution is limited by the optics (wavelength x0.61/NA ~200 nm), the pixel resolution (this depends on the pixel size of the cameras, not clearly described) and the details of image processing/analysis. Roughly speaking, the spatial frequency of a loose actin network would be 50~500 nm and is roughly at the same order with the imaging resolution. Thus, the slight decrease of the anisotropy detected in Figure 1 might be able to be explained as a consequence of the uniform shrinkage of an isotropic network instead of the alignment of the filaments. A good theoretical argument or an experimental demonstration against this possibility should be necessary.

3) Possible influences of the local geometry of the cell surface and the thickness of the cortical actin network on the anisotropy measurement have not been sufficiently considered/discussed. Is the assumption that the cortical actin structure is a smooth sheet without thickness realistic? The enrichment of microvilli in the cleavage furrow has been reported in mammalian cells and sea urchin embryos (eg. Yonemera 1993 https://www.ncbi.nlm.nih.gov/pubmed/8421057 and http://gvondassow.com/Research_Site/Picture_of_the_week/Entries/2010/2/23_Microvilli_on_the_sea_urchin_zygote.html). In the author's own data, the actin cortex seems thicker in the furrow (e.g. Figure 2). How uniform is the anisotropy along the direction of the thickness (i.e. perpendicular to the cortex)?

4) Related to the above point, a major issue is that not even a single pair (of 0 and 90 degree polarization, or a quadruplet of 0, 45, 90 and 135 degree polarization) of the actual images of dividing cells captured by their fluorescence polarization scopes has been shown. With the LC–PolScope, the anisotropy should be able to be calculated for each camera pixel. On the other hand, it is not obvious whether the averaged value for a 2 µm x 2 µm area, which include significant amounts of the extracellular space and the non–cortical cytoplasm, is an appropriate descriptor of the cortical nisotropy. Representative examples of the actual images of both the fixed and live cells (as they are obtained with different microscope setups and analyzed in different ways) should be presented. For the fixed cells observed by the LC–PolScope, for example, both the pseudo–colored raw polarity images as in Figure 1 and the calculated anisotropy images, which must have been made for drawing the Figure 1 graphs, should be added in the main figure. Individual grayscale images of different polarity angles should also be shown as figure supplements. For the live observation, examples for the central and the bottom z–sections with or without blebbistatin treatment in time course should be presented.

5) The live anisotropy imaging of the central z–sections (Figure 1 and Figure 2) was performed using a set–up with two linear emission polarizers and analyzed along the outlines of the cells through a lengthy and complex procedure of image analysis, including watershed segmentation and B–spline fitting. A subtle difference in these processes can have a significant influence on the final anisotropy values. More detailed information should be provided, i.e., examples of each steps of image processing, the outline determined by the segmentation, a curve with the anchoring points obtained by the spline–fitting and the angles of the cortical sheet etc. overlaid on the close–up of a furrow, probably as figure supplements. In relation to the above point 3, the lack of the drop of the anisotropy after blebbistatin treatment might be a consequence of the lack of deep ingression and its influence on the calculation of the anisotropy. A proper control to separate biochemical/biophysical effects from the geometrical/image analysis effects would be necessary (e.g. artificial deformation of a metaphase cell). Or, does the blebbistatin show the same effect on the equatorial actin structure at the bottom of the cell observed in the same setup as in Figure 3 and Figure 4?

6) The quantification of the fluorescence anisotropy at the bottom section of the cells (Figure 3 and Figure 4) was done by a ratiometry between the two images obtained with perpendicularly (0 and 90 degrees) placed linear emission polarizers. By this way, however, the anisotropy in the 45 or 135 degree direction can't be detected. This can easily be understood by imagining a long filament bundle placed at the 45 degree angle, for example. It would look exactly identical in the two images acquired with 0 and 90 degree polarizations, respectively (irrespective of the relative angle of the fluorophore's dipole moment). This highly anisotropic structure can't be distinguished from a uniform isotropic object such as fluorescent plastic with random fluorescence dipole orientation! Actually, "Ratio H/V" in Figure 1—figure supplement 4D is equal to 1 every 90 degrees at 45, 135, 225 and 315 degrees (this also indicates that the axes of polarization are at 0 and 90 degree angles relative to the camera).

The information about the anisotropy in the 45 and 135 degree directions lost at the image acquisition can't be compensated for by post–imaging computational analysis. Indeed, in Fishkind and Wang (1993), the spindle axis of each cell was carefully oriented parallel or perpendicular to the angles of polarization by using a rotatable stage. However, I couldn't find any description about the relative orientation between the angles of polarization and the cell division axis in this manuscript. Was it random (as Figure 1—figure supplement 4D suggests) or adjusted to a fixed angle?

7) Fishkind and Wang (1993) calculated (F_parallel – F_perpendicular)/F_parallel + F_perpendicular) (here, F_ denotes the fluorescence intensity) as a parameter for a preferential orientation of actin filaments instead of the simple ratio, F_parallel/F_perpendicular, which is used in this manuscript. What is the rationale for the simple ratio?

In formula (9), p_iso, which was defined in the formula, is a natural extension of the above formula by Fishkind and Wang while r_corrected is derived from the simple ratio (formula (3)). How could these values be linearly correlated as in formula (9)? What's the rationale for this formula?

8) The above 45 and 135 degree angle issue (point 6) is also relevant for the live imaging of the central z–sections (Figure 1, Figure 2). The regression with metaphase cortices as a model (Figure 1—figure supplement 4D) is effective for rectifying the over– or under–estimated anisotropy due to the orientation of the cortex. However, it is of no use for restoring the lost anisotropy along the 45 or 135 degree directions that might arise independently of the orientation of the cortex but lost at the acquisition.

9) Release or expulsion of filaments from the contractile ring has been reported in fission yeast and HeLa cells (https://elifesciences.org/content/5/e21383, http://www.nature.com/articles/ncomms11860). Could a similar process be a simpler explanation for the observed decrease of the fluorescence anisotropy at the furrow in the central z–sections (Figure 1)? Actually, filaments roughly perpendicular to the furrow cortex are detected in the Center Section/Late Ingression panel of Figure 3. This would also be able to account for the lack of the drop in the anisotropy after the blebbistatin treatment (Figure 2).

10) The displacement of the cortex after laser surgery is primarily caused by a viscoelastic response of the remaining cortical network. After the perpendicular cut (Figure 4), there remained almost no flanking actin structure that linked the upper and lower sides of the cut. On the other hand, after the parallel cut (Figure 4), there remained massive actin structures next to the lesion. These remaining links between the left and right sides of the lesion might resist against the deformation. Even if the tension was totally isometric, the same results might be observed. In a word, the comparison is not fair. What happens if a circular lesion sufficiently smaller than the width of the ring is made? Does the hole expand anisotropically? Conversely, what would happen if the 'Parallel cut' is made across the entire length of the equator?

*Reviewer #3:*

The paper by Gerlich and colleagues investigates an important question pertaining to how actin filaments are organized during cytokinesis in cultured mammalian cells. The findings described are interesting, but not totally novel. This is due to the fact that essentially similar findings have been reported by Fishkind and Wang in a seminal paper in JCB in 1993. However, this itself shouldn't undermine the work of Gerlich and colleagues, since they use better time–resolved approaches to re–investigate this important question. I also believe it is important to revisit old classic models with improved technology.

In general, I like the study, but I thought a simple experiment that could have been done is to carry out the laser cutting experiments in low doses of paranitroblebbistatin or jasplakinolide to determine if the tension generated in the aligned filaments is dependent on myosin II or actin disassembly. I realize the authors have shown that the transition to an anisotropic state depends on myosin II activity, but this is done with high doses of pn–blebbistatin. This simple experiment will provide a functional and molecular touch to the filament organization they describe.

[Editors’ note: what now follows is the decision letter after the authors submitted for further consideration.]

Thank you for resubmitting your work entitled "Actomyosin contracts as a randomly oriented network to initiate cytokinesis in vertebrate cells" for further consideration at *eLife*. Your revised article has been favorably evaluated by Anna Akhmanova (Senior editor) and three reviewers, one of whom, Mohan Balasubramanian is a member of our Board of Reviewing Editors.

The manuscript has been improved but there are a large number of remaining issues that need to be addressed before acceptance, as outlined below.

Please note that even though these concerns are all potentially fixed with some rewriting, it is nonetheless essential that you pay full attention to them and address them. A final decision will be made by the editors and will depend crucially on your responses and modifications to the text.

Essential revisions:

1) The filament sliding model is mentioned in the Introduction as a possible model and the suggestion that the cytokinetic ring contraction might be driven by myosin sliding anti–parallel actin filaments, but the authors don't get back to these ideas in the discussion to disseminate whether their work has helped to shed light on these issues.

2) The authors also fail to discuss their findings with respect to the work of Fishkind and Wang, 1993 which already indicates network alignment in the early anaphase (Figure 6 in Fishkind and Wang, 1993).

3) The claim by the authors that the data indicates that 'cytokinetic cleavage furrow ingression initiates by contraction of a randomly oriented actomyosin network' is misleading as it indicates that a global network contraction would lead to the localized furrow ingression. And the authors and others have shown that locally increased myosin activity is most likely the driving factor (Wollrab et al., 2016). The present data does not show how the contraction of a randomly oriented actin network would lead to the furrow ingression without alignment of filaments.

4) It is not clear how the authors compute the normalized polarization factor. Neither in the results part nor in the Materials and methods section is the formula mentioned.

5) In connection to this are also some comments missing about the precision of the polarization experiments: given the huge spread in the normalization of the polarization factor for aligned actin filaments (0.75–1.5!!!), how sensitive will the method be for changes in the cortex alignment. I.e. how many filaments in the cortex have to be aligned so that the polarization factor differs significantly from the cortex value?

6) Subsection “Contraction of a randomly oriented actin network initiates cleavage furrow ingression”: Given the rather limited time resolution and sensitivity of the polarization measurements, the last statement should be made more carefully (especially as later measurements with TIRF do indicate an earlier filament alignment).

7) Subsection “Contraction of a randomly oriented actin network initiates cleavage furrow ingression”: I don't think that the authors referenced correctly here. I would expect to see: Chugh, P, G. Charras, G. Salbreux, and E.K. Paluch. 2017. Nat. Cell Biol. 19: 689–697. Also, later in the Discussion, as this work studies cortex tension and cytoskeletal architecture.

8) Subsection “Equatorial actin filament reorientation depends on myosin II activity”: It is interesting to see that actin accumulation at the equator is not myosin dependent. But it might be worth a word to mention that the cortex thickness seems not to be altered despite a higher actin content. This indicates again that the actin is aligned.

9) Subsection “Equatorial actin filament reorientation depends on myosin II activity”: In the context of myosin driven actin accumulation, the work of Wollrab et al. should be discussed here as they analysed in detail the recruitment and activity of myosin during ring contraction (Wollrab et al., 2016).

10) Subsection “Equatorial actin filaments reorient on planar cortex regions”: Important, now the authors stated that they detect alignment together with the onset of furrow ingression. It would be good to point out that the improved spatial resolution increased the sensitivity for changes in filament alignment and in contrast to the earlier experiments where they were not able to detect any changes in alignment at such early time points.

11) Subsection “Cortical tension is highly directional at the cell equator”: How did the authors look at cortical flows? What is their definition of long vs short range?

12) Please see figure comments regarding the micro–surgery experiments.

13) Discussion section: The discussion is very hand waving and needs thorough rewriting. It seems that out of nothing, the authors say now that everything is driven by local myosin activity. This is well possible, but then the observed network alignment is not a big deal, and the question remains how the myosin activity is localized to the ring. The authors fail to put their work in context with earlier work, such as Fishkind and Wang, how the experimental improvements help in understanding the network organization during cytokinesis, and how their findings will help to solve the ongoing debate about the most conclusive model.

14) “Our study shows that disordered actomyosin networks can generate sufficient mechanical tension to initiate cleavage furrow ingression in vertebrate cells”: The authors do not test whether the disordered actomyosin network generates enough tension for furrow ingression. Especially the blebbistatin experiments show that there are signals fostering actin filament polymerization at the furrow site without ingression. To support the above statement, one would have to do a laser ablation next to the developing furrow ingression and observe, if and how the ingression evolves further or not.

15) “This suggests that vertebrate cells align the actomyosin network

directly at the site of highest contractility, whereby lateral cortical movements adjacent to the cell equator”: It's not very clear what the authors want to point out here. Cells do not align the actomyosin at the site of contractility, contractility itself aligns generally the actin filaments.

16) “Given that actomyosin rings mediate various other biological processes, including cellular wound healing and developmental morphogenesis, it will be interesting to further investigate the intricate relationship between contractile forces and actin network organization”: Very generic statement. Can be left out.

17) Major point 1: Interpretation of the low anisotropy regions in the mid–plane

The metaphase cell in Figure 1 shows a clear trend that the regions with strong signals in the average image show lower color saturation in the orientation map (white), hence, lower polarisation. This backs up my arguments about the scaling effect (major point 2 in my previous comments), i.e., denser the structure, lower the calculated anisotropy. Apart from the theory, the authors should provide good reasoning for the interpretation of "white" signal at the furrow as perpendicular alignment while those in the metaphase cell are clearly not representing such alignment.

18) Major point 2: Comparison with a previous report

The polarization microscope setups used in this study are most straight–forward and sensitive in detection of the polarization within the X–Y plane. Indeed, the pioneering work with a similar setup by Fishkind and Wang focused on this. Honestly, I don't understand why the authors insist on not taking this approach (e.g. flatten a cell in a way similar to Figure 3) on their LC–Polscope. This should allow straight–forward visualization of the filament alignment and direct comparison with the results by Fishkind and Wang. Anyway, Figure 3 data, in principle, provide a live version of Fishkind and Wang, which is indeed a drastic improvement. Similarities, differences and improvements should be properly discussed in Discussion section, including the difference in the data analyses (simple ratiometry in Figure 3 vs (F_parallel – F_perpendicular)/(F_parallel + F_perpendicular in Fishkind and Wang).

19) Major point 3: Laser surgery in Figure 4

As I pointed out as Major point #10 in my previous comments, the comparison between "perpendicular" vs "parallel" cuts is not fair. Why a smaller round hole was not tried or is not suitable should be explained.

20) Subsection “Contraction of a randomly oriented actin network initiates cleavage furrow ingression”.

As a sample with isotropic fluorescence dipole distribution, we imaged fluorescent plastic (Figure 1). We then normalized the polarization factor to be in a range between 1 and 0 for maximally aligned fluorophores (stress fibres) and isotropic plastic, respectively.

I expect the isotropic plastic to have a polarisation factor of 0.5 much akin to the metaphase cortex. Is there something specific about the substrate?

21) Figure 1.

During late stages of anaphase, the normalised polarisation at the poles appear to be more than 0.6 (mean) in Figure 1 and less than 0.6 in Figure 1. Can the authors explain the discrepancy?

22) Figure 1.

The authors seem to let go of a major point by not comparing the normalised potential of either equator or poles for different time points. This would have made a nice point regarding orientation of actin over time.

23) Laser Microsurgery experiment:

This is indeed a very interesting observation. The cut in the parallel direction on the cell cortex doesn't induce cortex movement which is in stark contrast to a similar cut made in the perpendicular orientation. It will however be interesting to see whether the parallel slit can induce outward movement if its horizontal dimension (which is a line currently) is increased. If so, then does this minimum lateral dimension (when outward motion is achieved) correspond to the FWHM of the anisotropic actin network measured in the paper (4.4.um)?

24) As a general interest it would be interesting to know, what happens to these cells after the cut is made. Do these cells show a healing response or abort cytokinesis and undergo cell death.

Issues with figures

Figure 1: The question marks can be left out.

Figure 1) The authors might want to discuss why the cortical actin shows parallel alignment along the plasma membrane due to the overall confinement of the actin filaments along the membrane. And that the white area along the furrow indicates the alignment of filaments along the z axis. Since this kind of measurements are not common for the major audience, a careful description and discussion are important.

Figure 1) Given the spread of the data, tracing the difference of individual cells would be helpful by connecting the corresponding red and blue dots.

Figure 2: The cell cortex appears jagged and the 0 and 90 degree channels do not seem always properly aligned (very clear at 330sec). Why is that the case? Channel alignment is essential for the polarization analysis.

Figure 2—figure supplement 1: It would be instructive if the authors could first show the raw intensities before computing the ratio.

Figure 2—figure supplement 1:Hhow can you obtain negative values for the actin cortex thickness?

Figure 3: The plot indicates that changes in network orientation appear at about 100sec. this would mean that alignment would go hand in hand with the furrow ingression. The authors should discuss this point as it is in conflict with the earlier conclusion that alignment can be only detected at later time points.

Figure 4: The authors should show a bleach control to show that their ablation protocol really cuts the actin network. They also should provide data to indicate whether the plasma membrane stays intact or not during the procedure.

In addition, it would be very instructive for the authors conclusion to perform a laser ablation of the network adjacent to the cytokinetic ring in the beginning of furrow ingression and to observe whether the contraction progresses or not. If the global contraction of the random actin network is necessary for the cytokinetic ring, then this ablation should perturb the process.

Figure 5: The schematic looks nice, but is not very instructive. In addition, the model is not properly formulated (i.e. in a theoretical way) and the claims are not presented in clear connection to the presented data. How is the equatorial contractility generated in the first place (–> increased local myosin activity)? how to explain lateral cortex displacement without increase of tension (–> less connectivity in the network? Shorter filaments?) How is the higher tension along the equator generated (–>via filament alignment)?

Figure 1—figure supplement 1.

The area for selected at the poles is mentioned as 16um in the figure against 10um in the text (Materials and methods section).

Figure 2

Curvature change in Figure 2 appears to reduce post 350 second. No such trend however is observed for Figure 2 for essentially the same measurement.

Figure 2 and Figure 1.

In Figure 2, there appears to be no net change in normalised emission ratio, indicative of polarisation, at the poles. However, in Figure 1, there seems to be a modest increase in normalised polarisation.

Figure 4.

It would be interesting to know whether the increasing movement of the network on ablation scales with increased actin alignment.

---

## [Author Response]

[Editors’ note: the author responses to the first round of peer review follow.]

[…] Reviewer #1:Spira et al., present a valuable new approach to spatiotemporally monitor changes in actin filament organization during cleavage furrow ingression. Previously, EM and quantitative fluorescence microscopy–based assays have visualized the formation of discreet foci of actin filaments prior to cleavage furrow formation. The method proposed here, fluorescence polarization microscopy, provides an opportunity to monitor dynamics of actin filament organization during cleavage furrow ingression and link it directly to ring contraction in a live setting. Based on their work the authors conclude that the, (a) actin reorientation appears not to be critical for transducing forces leading to the early stages of cleavage furrow ingression, (b) actin filament reorientation is a consequence of mechanical input from myosin II, and (c) filament reorientation is a consequence of the mechanical coupling along the contractile ring. However I have a few concerns that I would like the authors to address before publication.The results presented in this manuscript are largely consistent with those observed in other cellular systems such as sea urchin and fission yeast, namely that filament reorganization occurs much later during cleavage furrow ingression and is not necessarily a cause for the initiation of furrow formation which reduces the novelty of the authors' conclusions.That the actin network transforms from a largely isotropic cortical meshwork in metaphase cells into an anisotropic network shortly after the onset of anaphase onset has already been reported earlier (see Fishkind and Wang, 1993). Understandably, the present work provides important dynamic insights along with data to suggest presence of radial forces along the cell equator that drive ring contraction and as such is interesting.However the authors should clarify the following concerns before publication. In general, the description of their methods needs better articulation. In particular I would like the authors to clarify on the following points:

We thank the reviewer for appreciating our experimental approach and the biological significance of our findings. We are also grateful for pointing out that the methods description should be improved. We have addressed all concerns by extending the documentation of fixed and live cell data in three new supplemental figures (new Figure 1—figure supplement 1 and Figure 2—figure supplement 1 and Figure 2—figure supplement 2) and by extensive rewriting of the Results and Materials and methods sections, as explained below.

Definition of polarization factor p is not clear. Is it the same as anisotropy?

The terminology was indeed not well explained in the original manuscript. The polarization factor expresses the degree of actin network anisotropy measured by LC-PolScope microscopy, and it is calculated by normalizing the measured fluorophore anisotropy to reference values for fluorescent plastic (polarization factor = 0) and AlexaFluor 488-phalloidin-stained stress fibers (polarization factor = 1). This is now stated in the main text and explained in detail in the Materials and methods section.

What is r_norm in formula (9)? Is it normalization polarization factor? It is very hard to follow since the description of methods is poorly written.

The r norm in formula (9) indicates the normalized emission ratio, which we calculated to measure actin network organization in live mitotic cells with the live-cell confocal set-up. The live-cell confocal set-up excites the probe with the linearly-polarized laser aligned with the optical table and sequentially records two images with perpendicularly-oriented polarization filters in the emission path, horizontal (H) and vertical (V) relative to the optical table. We used this microscope for imaging of cleavage furrow ingression in live cells, as the LC-PolScope acquired images at a substantially slower rate, resulting in motion artefacts.

To clarify the measurement procedures, we included raw images and intermediate processing results to revised Figure 1 and Figure 2 and added supplementary figures illustrating the analysis procedure for both microscopy set-ups. We use the term normalized polarization factor to interpret alignment of actin filaments using the LC-PolScope setup and normalized emission ratio for images acquired with linear emission polarizers in the live-cell set-up.

According to the definition of anisotropy (subsection “Quantification of fluorophore alignment with Confocal LC-PolScope”), why would a random network have 50% anisotropy (Figure 1)? I would expect I_0 = 1, I_90 = 0, I_45 = 0.5, I_135 = 0.5 and hence a = 1, b = 0, c = 2 and therefore anisotropy = 1.

The analysis methodology and rationale behind the predicted network polarization values was indeed not well explained in the original manuscript. We hypothesized that in a 200 nm-thin metaphase cell cortex, actin filaments are predominantly aligned with the plane of the cell surface, but oriented in random directions parallel to the cell surface. If this model were correct, then this should yield a normalized polarization factor (see above for definition) of 0.5 when the cell cortex is imaged in an orientation perpendicular to the focal plane, based on the mixture of fluorophore dipoles aligned with the focal and cortical plane, as well as dipoles that orient perpendicular to focal plane. We now specify the assumptions about metaphase cortex organization that would predict a polarization factor of 0.5 and present this as a model to be tested by experiments. We further provide extensive illustration of intermediate processing results and rephrased the main text and Materials and methods section for clarification.

Figure 1: Early furrow ingression, the polarization factor varies in a huge range. It is therefore unreliable to say the network is still random.

We agree that the high variability and low sample number limited the interpretation of the original data set. The variability in the data is likely attributed to both imaging noise and biological variation. To improve the accuracy of median estimation, we acquired and analysed 80 additional cells (revised Figure 1). Moreover, we binned the data in 4 instead of 3 time windows to achieve higher temporal resolution. The extended data analysis confirms that during early furrow ingression (101-150 s) the network contains randomly oriented actin filaments. These findings are also consistent with our live cell fluorescence ratio measurements shown in Figure 1. Overall, the extended data corroborates the conclusion that cleavage furrow ingression initiates by contraction of a randomly oriented actin network.

Figure 1: looks like anisotropy starts to reduce at 60 sec while ingression starts at ~ 90 sec (Figure 1). So it does not look like "Contraction of an isotropic actin network initiates cleavage furrow ingression" as claimed.

We thank the reviewer for pointing out potential ambiguity of data interpretation by visual inspection. We have therefore performed additional analyses and statistical testing to determine the onsets of actin cytoskeleton re-orientation and cleavage furrow ingression. To objectively measure cleavage furrow ingression, we applied circular fits to cortex at centre position between the segregating chromosome masses (new Figure 2). We then performed statistical testing comparing cortex curvatures of different time points to determine the onset of cleavage furrow ingression. We also performed statistical testing comparing fluorescence anisotropy between different time points to determine the onset of actin network alignment (new Figure 2). These new data show that furrow ingression initiates at 90 s, while significant actin network only occurs at 120 s after anaphase onset. These measurements corroborate that a randomly oriented actin network initiates cleavage furrow ingression.

Discussion section, "Our study shows that disordered actomyosin networks can generate sufficient mechanical tension to initiate cleavage furrow ingression in vertebrate cells": This is of low confidence as mentioned above.

We agree that the low amount of sample points in the original manuscript was suboptimal. However, our extensive new data and analyses (see response above) provide strong support for this important statement.

Subsection “Equatorial actin filaments reorient on planar cortex regions”: "While these data do not rule out that local furrowing facilitates actin filament alignment, they show that planar network contraction is sufficient to induce substantial alignment." Comment: the network is not only simply planar, but also part of the ring. Alignment would occur along the ring (and this is expected), including the part on the planar network.

This was indeed phrased in a misleading way. We intended to refer to *local* curvature, which was proposed to be relevant for inward-directed network movement towards the leading edge of the cleavage furrow (Dorn et al., 2016). We thank the reviewer for pointing out that this was unclear. We think that this statement does not add much to the interpretation of our data and therefore removed it from the revised manuscript.

Figure 4, post–cut time should be reported to tell whether cells were imaged at the same post–cut time.

We thank the reviewer for pointing out that the information on post-cut time was missing. The postcut time point is 0.9 s for all images, as indicated in the revised Figure 4.

"Mechanical coupling" is mentioned at a few places, but what does it mean? What is coupled to what?

By the term “mechanical coupling”, we intended to highlight the fact that the contractile cortical zone is continuous along the equatorial cell circumference, as opposed to the orientation perpendicular to the equatorial cortex, where the contractile cortex is adjacent to low contractility cortical regions. To improve clarity, we rephrased the respective sections of the manuscript to avoid this term.

Title: "Actomyosin contracts as a network to drive cytokinesis in vertebrate cells". Comment: The data did not convince if the whole network contracts or part of it forms a ring which then contracts.

Our extensive new data corroborate that cleavage furrow ingression initiates by contraction of a randomly oriented network. At later cytokinesis stages, the degree of actin filament alignment remains relatively low, yet we cannot exclude that a small fraction of filaments forms a ring with parallel filaments. To improve the accuracy of the title, we rephrased: “Actomyosin contracts as a randomly oriented network to initiate cytokinesis in vertebrate cells”.

Reviewer #2:Animal cells execute cytokinesis by ingression of the cleavage furrow that is driven by contraction of the ring–like actomyosin network at the cell equator. The details of the dynamic organization of actin filaments in the ring are not well understood. The authors employed fluorescence polarization microscopy to determine the anisotropy of actin filaments in fixed and live cells, and detected gradual alignment of actin filaments during furrow deepening. They also performed laser microsurgery experiments and discovered an anisotropy in the cortical mechanical tension at the cell equator.Although Fishkind and Wang, 1993 already demonstrated an increase of anisotropy of actin filaments at the equatorial cortex during cytokinesis of mammalian cultured cells using fluorescence polarization microscopy in the 90's, it was in fixed cells with a non–confocal set–up and thus the z– and time–resolution was not high. It is of significant interest to revisit this issue using modern microscopy technology, which has progressed in the last two decades. Indeed, the images of fixed and live interphase cells by a confocal LC–PolScope (Figure 1) are amazing. However, unfortunately, there are many points to be clarified or improved as below, which would prevent publication of this interesting work in the current form. The experimental evidence doesn't seem to be sufficient for discussions with physical terms such as "dissipation" or" force feed–back".

We are delighted that the reviewer appreciates our technical approaches and the importance of our findings. We are grateful for the thoughtful comments on where the manuscript needs clarification and the detailed suggestions how to improve the analysis and discussion of our data. We have added extensive new data to better illustrate the analysis procedures and have rephrased the discussion along the lines suggested by the reviewer, as explained in detail below.

1) Fluorescence polarization microscopy is more powerful for the anisotropy in the X–Y plane than that in a plane parallel to the Z–plane. Indeed, Fishkind and Wang (1993) examined the anisotropy in the X–Y plane in a similar scheme to that used for Figure 3 and 4. Why wasn't the confocal LC–PolScope used for the observation of the bottom of the cell (after fixation if necessary)? How significant is the motion artifacts in dividing cells?

Acquisition of a single optical section with the LC-PolScope requires recording of 5 image frames at different polarization angles. Our LC-PolScope implementation enabled to record at a frame rate of 5 s per channel. The >20 s time-lapse between recording the first and last polarization frame resulted in a substantial cortex displacement during cleavage furrow ingression in live cells, which we could not compensate by image registration. For the study of actin organization in live anaphase cells, we hence had to rely on the live-cell set-up with two perpendicularly-oriented polarization filters in the emission path, which was much faster as it did not require synchronization between microscope-controlling software and the LC controller.

Regarding the imaging of flat surfaces, we have now included original image data showing live-cell images of the bottom surface in confined anaphase cells (Figure 3). We still prefer to use the centre sections for quantifications in most other experiments, as we can then apply accurate cortex segmentation excluding filopodia from the analysis. Moreover, the cell confiner is not compatible with blebbistatin treatment (see point reviewer 2 point 5).

2) In general, the anisotropy measurement depends on the relative scale between the spatial frequency of the target structure and the spatial resolution of the imaging. Imagine an infinitely repeating orthogonal grid of 100 µm interval. If the measurement is done at the scale of 1 mm or larger, the anisotropy of this structure would be uniformly 0. On the other hand, at the resolution of 10 µm, the anisotropy would be non–uniform, i.e., >0 on a line while 0 between the lines. This would result in >0 value after spatial averaging. This means that, even for the same structure, a smaller anisotropy value would be obtained when the spatial resolution becomes lower. Conversely, with a constant spatial resolution, isotropic shrinkage of a network structure would result in a smaller anisotropy value.In the actual measurement of the cortical actin network, the distribution of the spatial frequencies is complex and can range from ~7 nm (the diameter of a single actin filament), 10~100 nm (filament–to–filament distance in a bundle such as a microvillus, a sarcomere or a stress fiber), or bigger (~µm). The imaging resolution is limited by the optics (wavelength x0.61/NA ~200 nm), the pixel resolution (this depends on the pixel size of the cameras, not clearly described) and the details of image processing/analysis. Roughly speaking, the spatial frequency of a loose actin network would be 50~500 nm and is roughly at the same order with the imaging resolution. Thus, the slight decrease of the anisotropy detected in Figure 1 might be able to be explained as a consequence of the uniform shrinkage of an isotropic network instead of the alignment of the filaments. A good theoretical argument or an experimental demonstration against this possibility should be necessary.

We thank the reviewer for pointing to these potential technical limitations, which we had not discussed in our original manuscript. Our analysis computes the degree and orientation of actin filament alignment within an ROI of 2 µm length and 0.6 µm width. The dominant orientation and the polarization factor are moments of the angular distribution of dipoles over the analysed region. The dominant orientation is the zeroth angular moment (mean) and the polarization factor is the first angular moment (standard deviation) around the mean. If the actin network is assumed to have the highest spatial period of 200 nm (matching the resolution limit of optics), shrinkage of uniform network will not affect measured parameters as the analysis window of 2 µm is an order of magnitude larger. If the actin network is assumed to have even higher spatial period (e.g., 500 nm), we would resolve them. All our data is acquired with Nyquist sampling and we detected the cortex as a continuous stretch of intensity, thus spatial period between filaments in our cells is finer than 200 nm. Hence, the detected changes in fluorescence anisotropy report on net filament orientation within the ROI rather than network contraction.

3) Possible influences of the local geometry of the cell surface and the thickness of the cortical actin network on the anisotropy measurement have not been sufficiently considered/discussed. Is the assumption that the cortical actin structure is a smooth sheet without thickness realistic? The enrichment of microvilli in the cleavage furrow has been reported in mammalian cells and sea urchin embryos (eg. Yonemera 1993 https://www.ncbi.nlm.nih.gov/pubmed/8421057 and http://gvondassow.com/Research_Site/Picture_of_the_week/Entries/2010/2/23_Microvilli_on_the_sea_urchin_zygote.html). In the author's own data, the actin cortex seems thicker in the furrow (e.g. Figure 2). How uniform is the anisotropy along the direction of the thickness (i.e. perpendicular to the cortex)?

The local geometry of the cell surface and the thickness of the cortical actin network are indeed important aspects that need to be considered to correctly interpret anisotropy measurements. We did indeed exclude microvilli protruding from cortex regions by segmenting thin cortical contours for anisotropy measurements. In the original manuscript, we had illustrated the measurement regions by squares around the actual segmented cortex regions to avoid masking the original image data. We recognize based on the reviewer’s concern that this was misleading, and therefore now indicate the polar and equatorial measurement regions by arrowheads in the main figure and show the actual segmentation masks side-by-side the anisotropy images in Figure 1—figure supplement 1.

We thank the reviewer for pointing out that changes in cortical thickness might also affect the anisotropy measurements. We had not considered this in our original analysis and hence performed new experiments to measure the cortical thickness based on relative positions of a plasmamembrane-associated marker and an actin marker, as reported in Clark et al., 2013. We measured a cortical thickness of ~200 nm in metaphase cells, which was similar in furrow regions of anaphase cells (new Figure 2—figure supplement 1 G-J). We did detect a significantly thicker cortex in interphase cells as previously reported (Chugh et al., 2017), validating the sensitivity of our measurement procedure. Overall, these data show that the observed changes in fluorescence anisotropies during cytokinesis progression cannot be attributed to changes in cortical thickness.

The reviewer further suggests measuring fluorescence anisotropies along line profiles perpendicular to the cell cortex. As the cortical thickness is below the resolution limit of our microscope, such measurements will not be informative. However, we think that the new data and improved illustration of image segmentation clarify that the detected fluorescence anisotropy changes are not due to changes in cortical thickness or local geometry effects.

4) Related to the above point, a major issue is that not even a single pair (of 0 and 90 degree polarization, or a quadruplet of 0, 45, 90 and 135 degree polarization) of the actual images of dividing cells captured by their fluorescence polarization scopes has been shown. With the LC–PolScope, the anisotropy should be able to be calculated for each camera pixel. On the other hand, it is not obvious whether the averaged value for a 2 µm x 2 µm area, which include significant amounts of the extracellular space and the non–cortical cytoplasm, is an appropriate descriptor of the cortical nisotropy. Representative examples of the actual images of both the fixed and live cells (as they are obtained with different microscope setups and analyzed in different ways) should be presented. For the fixed cells observed by the LC–PolScope, for example, both the pseudo–colored raw polarity images as in Figure 1 and the calculated anisotropy images, which must have been made for drawing the Figure 1 graphs, should be added in the main figure. Individual grayscale images of different polarity angles should also be shown as figure supplements. For the live observation, examples for the central and the bottom z–sections with or without blebbistatin treatment in time course should be presented.

We thank the reviewer for pointing out the incomplete documentation of raw and processed image data in our original manuscript. In the revised manuscript, we now show representative original images from the LC-PolScope (new Figure 1 and new Figure 1—figure supplement 1), as well as for the live-cell polarization filter setup (new Figure 2, Figure 2—figure supplement 2, and Figure 3). We also show the pseudo-colored raw polarity images for cells in cytokinesis (new Figure 1) and the raw polarity images as calculated for Figure 1 (new Figure 1—figure supplement 1).

Regarding the comment on the squared ROI, we would like to refer to the response to point #3. The graphical illustration of squares in the original figures was misleading, as the actual anisotropy measurements were performed within thin cortical ROIs fitted precisely to the cell contour (new Figure 1—figure supplement 1).

5) The live anisotropy imaging of the central z–sections (Figure 1 and Figure 2) was performed using a set–up with two linear emission polarizers and analyzed along the outlines of the cells through a lengthy and complex procedure of image analysis, including watershed segmentation and B–spline fitting. A subtle difference in these processes can have a significant influence on the final anisotropy values. More detailed information should be provided, i.e., examples of each steps of image processing, the outline determined by the segmentation, a curve with the anchoring points obtained by the spline–fitting and the angles of the cortical sheet etc. overlaid on the close–up of a furrow, probably as figure supplements. In relation to the above point 3, the lack of the drop of the anisotropy after blebbistatin treatment might be a consequence of the lack of deep ingression and its influence on the calculation of the anisotropy. A proper control to separate biochemical/biophysical effects from the geometrical/image analysis effects would be necessary (e.g. artificial deformation of a metaphase cell). Or, does the blebbistatin show the same effect on the equatorial actin structure at the bottom of the cell observed in the same setup as in Figure 3 and Figure 4?

We thank for the suggestions how to improve the illustration of the image analysis procedure. We have included examples for each step of the image processing pipeline in the revised manuscript (new Figure 2—figure supplement 2) and extended and rephrased the main text and Materials and methods section for clarification.

To address the concern on potential geometry effects, we established a new analysis pipeline to measure cortex curvature at different time points. On this basis, we selected time points in which the absolute curvature of the equatorial cell cortex does not exceed the initial curvature in metaphase (new Figure 2). We found that the polarization ratio significantly dropped at cortex curvatures that did not exceed those of metaphase cells (compare Figure 2). Hence, the suppression of fluorescence anisotropy changes by blebbistatin cannot be solely explained by geometry effects.

We appreciate the suggestion to image the bottom surface of blebbistatin-treated cells under confinement. However, this is technically not feasible as blebbistatin is inactive in the confiner (i.e., blebbistatin also does not inhibit lateral furrow ingression in the confiner). This is likely due to blebbistatin adsorption to the polymer of the confinement chamber, as has been observed for other chemical compounds.

6) The quantification of the fluorescence anisotropy at the bottom section of the cells (Figure 3 and Figure 4) was done by a ratiometry between the two images obtained with perpendicularly (0 and 90 degrees) placed linear emission polarizers. By this way, however, the anisotropy in the 45 or 135 degree direction can't be detected. This can easily be understood by imagining a long filament bundle placed at the 45 degree angle, for example. It would look exactly identical in the two images acquired with 0 and 90 degree polarizations, respectively (irrespective of the relative angle of the fluorophore's dipole moment). This highly anisotropic structure can't be distinguished from a uniform isotropic object such as fluorescent plastic with random fluorescence dipole orientation! Actually, "Ratio H/V" in Figure 1—figure supplement 4D is equal to 1 every 90 degrees at 45, 135, 225 and 315 degrees (this also indicates that the axes of polarization are at 0 and 90 degree angles relative to the camera).The information about the anisotropy in the 45 and 135 degree directions lost at the image acquisition can't be compensated for by post–imaging computational analysis. Indeed, in Fishkind and Wang (1993), the spindle axis of each cell was carefully oriented parallel or perpendicular to the angles of polarization by using a rotatable stage. However, I couldn't find any description about the relative orientation between the angles of polarization and the cell division axis in this manuscript. Was it random (as Figure Supplement 4D suggests) or adjusted to a fixed angle?

This is indeed an important technical concern, which we have taken into account by imaging only cells that had their spindle axis oriented along the linear emission polarizer filters, similar to Fishkind and Wang. We had selected only cells with a spindle axis oriented ± 20 degrees relative to X and Y axes. As this was described somewhat hidden in the Materials and methods section of the original manuscript, we extended the revised Materials and methods section and legend of the respective figures to clarify this concern.

7) Fishkind and Wang (1993) calculated (F_parallel – F_perpendicular)/F_parallel + F_perpendicular) (here, F_ denotes the fluorescence intensity) as a parameter for a preferential orientation of actin filaments instead of the simple ratio, F_parallel/F_perpendicular, which is used in this manuscript. What is the rationale for the simple ratio?In formula (9), p_iso, which was defined in the formula, is a natural extension of the above formula by Fishkind and Wang while r_corrected is derived from the simple ratio (formula (3)). How could these values be linearly correlated as in formula (9)? What's the rationale for this formula?

In contrast to Fishkind and Wang, we use a polarized excitation beam to excite fluorescent dipoles. This makes the calculation of anisotropy more difficult and we cannot use the formula reported in Fishkind and Wang (1993). Because of this limitation, we measured relative changes of actin filament orientation over time based on a simple ratio between the images acquired with the horizontal and the vertical emission polarizers. We extended and rewrote the Materials and methods section to clarify how we calculated anisotropies with the two different microscope set-ups used in our study.

8) The above 45 and 135 degree angle issue (point 6) is also relevant for the live imaging of the central z–sections (Figure 1, Figure 2). The regression with metaphase cortices as a model (Figure supplement 4D) is effective for rectifying the over– or under–estimated anisotropy due to the orientation of the cortex. However, it is of no use for restoring the lost anisotropy along the 45 or 135 degree directions that might arise independently of the orientation of the cortex but lost at the acquisition.

This is indeed a very important point that was not sufficiently discussed in the original manuscript. In these experiments, we also only imaged cells with a spindle axis of ± 20 degrees relative to the main X/Y axes of the image. This is now explained more clearly in the revised figure legend and extended Materials and methods section.

9) Release or expulsion of filaments from the contractile ring has been reported in fission yeast and HeLa cells (https://elifesciences.org/content/5/e21383, http://www.nature.com/articles/ncomms11860). Could a similar process be a simpler explanation for the observed decrease of the fluorescence anisotropy at the furrow in the central z–sections (Figure 1)? Actually, filaments roughly perpendicular to the furrow cortex are detected in the Center Section/Late Ingression panel of Figure 3. This would also be able to account for the lack of the drop in the anisotropy after the blebbistatin treatment (Figure 2).

As explained in response to #3 and #4, we calculated fluorescence anisotropy in thin ROIs fitted to the cell cortex contour (see new Figure 1—figure supplement 1 and Figure 2—figure supplement 2). Filaments protruding into the cytoplasm hence do not affect the detected anisotropies in these regions.

10) The displacement of the cortex after laser surgery is primarily caused by a viscoelastic response of the remaining cortical network. After the perpendicular cut (Figure 4), there remained almost no flanking actin structure that linked the upper and lower sides of the cut. On the other hand, after the parallel cut (Figure 4), there remained massive actin structures next to the lesion. These remaining links between the left and right sides of the lesion might resist against the deformation. Even if the tension was totally isometric, the same results might be observed. In a word, the comparison is not fair. What happens if a circular lesion sufficiently smaller than the width of the ring is made? Does the hole expand anisotropically? Conversely, what would happen if the 'Parallel cut' is made across the entire length of the equator?

We thank the reviewer for pointing out that the local enrichment of F-actin in regions surrounding the cut could limit cortex displacement. We revisited the original data and found no evidence of higher accumulation levels of F-actin in regions surrounding the cut in either of the cutting directions. Hence, we do think that the measured differences in initial outward movement reflect an asymmetric distribution of tension.

We had previously performed experiments with cutting path lengths across the entire width of the equator, but found that such large cortical disruption causes cell explosion. While we agree that cutting small rings would be interesting, we currently do not have the resources to perform this experiment, as the first author of this study, who has carried out all laser microsurgery experiments, has already left the laboratory. We are convinced that the current large data set of >80 laser microsurgery experiments clearly demonstrates the asymmetric distribution of cortical tension at the cell equator and hope that the reviewer agrees that the proposed experiment is not essential for publication.

Reviewer #3:The paper by Gerlich and colleagues investigates an important question pertaining to how actin filaments are organized during cytokinesis in cultured mammalian cells. The findings described are interesting, but not totally novel. This is due to the fact that essentially similar findings have been reported by Fishkind and Wang in a seminal paper in JCB in 1993. However, this itself shouldn't undermine the work of Gerlich and colleagues, since they use better time–resolved approaches to re–investigate this important question. I also believe it is important to revisit old classic models with improved technology.In general, I like the study, but I thought a simple experiment that could have been done is to carry out the laser cutting experiments in low doses of paranitroblebbistatin or jasplakinolide to determine if the tension generated in the aligned filaments is dependent on myosin II or actin disassembly. I realize the authors have shown that the transition to an anisotropic state depends on myosin II activity, but this is done with high doses of pn–blebbistatin. This simple experiment will provide a functional and molecular touch to the filament organization they describe.

We thank the reviewer for the positive evaluation of our manuscript. We appreciate the suggestion to perform laser cutting in the presence of low-dose para-nitroblebbistatin or jasplakinolide. Unfortunately, these experiments are technically not feasible, as laser cutting requires cell confinement and the cell confiner chambers are not compatible with drug treatment (i.e., blebbistatin does not suppress cleavage furrow ingression). This is likely due to adsorption of the chemical to the polymer used for construction of the confinement chamber, as noted for other chemical compounds.

[Editors' note: the author responses to the re-review follow.]

Essential revisions:1) The filament sliding model is mentioned in the Introduction as a possible model and the suggestion that the cytokinetic ring contraction might be driven by myosin sliding anti–parallel actin filaments, but the authors don't get back to these ideas in the Discussion to disseminate whether their work has helped to shed light on these issues.

We agree that it would be good to extend the Discussion and have revised it accordingly:

“Prior fluorescence polarization microscopy had shown preferential orientation of actin filaments along the cytokinetic cleavage furrow of vertebrate cells. […] The late onset and low degree of actin filament alignment within the equatorial cortex suggests a network contraction mechanism underlying cytokinetic cleavage furrow ingression in vertebrate cells, yet our data cannot rule out the possibility that at late cytokinesis stages some actin filaments form antiparallel arrays underneath a randomly oriented actin network.”

2) The authors also fail to discuss their findings with respect to the work of Fishkind and Wang, 1993 which already indicates network alignment in the early anaphase (Figure 6 in Fishkind and Wang, 1993).

We thank the reviewer for suggesting to discuss our findings in more depth, particularly regarding the relationship to the prior work by Fishkind and Wang. Figure 6 of the Fishkind and Wang study reports actin filament alignment at the bottom surface of strongly adherent NRK cells and classifies mitotic stages based on Hoechst-stained chromosomes without information on the cleavage furrow ingression state. The data hence do not provide information on whether actin filaments of the equatorial actomyosin cortex align before or after the onset of cleave furrow ingression. Moreover, Fishkind and Wang do not provide reference data from a structure with aligned actin filaments (e.g., stress fibers), and it hence remains unclear to which degree the filaments in equatorial actin network align. The authors of the prior study were also concerned about the limitation of their widefield set-up, which suffers from strong out-of-focus background noise. In the extended and revised Discussion section now comment on the technical innovations and put our findings in direct comparison to the Fishkind and Wang study:

“Prior fluorescence polarization microscopy had shown preferential orientation of actin filaments along the cytokinetic cleavage furrow of vertebrate cells. […] Yet, in combination with the fixed cell data, these experiments establish a detailed kinetic profile of actin network organization during cytokinesis progression.”

3) The claim by the authors that the data indicates that 'cytokinetic cleavage furrow ingression initiates by contraction of a randomly oriented actomyosin network' is misleading as it indicates that a global network contraction would lead to the localized furrow ingression. And the authors and others have shown that locally increased myosin activity is most likely the driving factor (Wollrab et al., 2016). The present data does not show how the contraction of a randomly oriented actin network would lead to the furrow ingression without alignment of filaments.

We thank the reviewer for pointing out that our model might be misconceived without specification of the equatorial position of the contractile network. We therefore rephrased the title for clarification: “Cytokinesis in vertebrate cells initiates by contraction of an equatorial actomyosin network composed of randomly oriented filaments”. We also rephrased Abstract, Introduction, and the legend of Figure 5 to specify the equatorial position of the contractile actomyosin network.

4) It is not clear how the authors compute the normalized polarization factor. Neither in the results part nor in the Materials and methods section is the formula mentioned.

This formula was indeed missing and we are grateful that the referee pointed this out. We have added the formula for the normalized polarization factor p_norm_ to the revised Materials and methods part.

5) In connection to this are also some comments missing about the precision of the polarization experiments: given the huge spread in the normalization of the polarization factor for aligned actin filaments (0.75–1.5!!!), how sensitive will the method be for changes in the cortex alignment. I.e. how many filaments in the cortex have to be aligned so that the polarization factor differs significantly from the cortex value?

The variability of individual polarization measurements is indeed high. The large sample number, however, enables an accurate estimation of the mean. To indicate the accuracy of the calculated means, we added bars indicating s.e.m. to the revised Figure 1 and comment on this in the main text: “The polarization measurements varied quite substantially between individual cells, yet owing to the relatively large sample sizes, we obtained accurate estimates of the mean for each of the respective stages and cellular positions (s.e.m. ranging between 0.01 – 0.03, Figure 1).”

6) Subsection “Contraction of a randomly oriented actin network initiates cleavage furrow ingression”: Given the rather limited time resolution and sensitivity of the polarization measurements, the last statement should be made more carefully (especially as later measurements with TIRF do indicate an earlier filament alignment).

To address this concern, we rephrased the concluding sentence of this paragraph: “Overall, these observations indicate that the orientation of actin filaments in the equatorial network driving initial stages of constriction is indistinguishable from the random orientation observed in the cortex of metaphase cells.”

7) Subsection “Contraction of a randomly oriented actin network initiates cleavage furrow ingression”: I don't think that the authors referenced correctly here. I would expect to see: Chugh, P., G. Charras, G. Salbreux, and E.K. Paluch. 2017. Nat. Cell Biol. 19: 689–697. Also, later in the Discussion, as this work studies cortex tension and cytoskeletal architecture.

This was a technical mistake, as we of course intended to cite Chugh et al. in l. 140. We thank the reviewer for pointing this out and corrected it in the revised manuscript. Regarding the comment on the Discussion section, it is not clear to us where we should add another reference to Chugh et al. If the reviewer provides specific instructions, we would be happy to revise the Discussion accordingly.

8) Subsection “Equatorial actin filament reorientation depends on myosin II activity”: It is interesting to see that actin accumulation at the equator is not myosin dependent. But it might be worth a word to mention that the cortex thickness seems not to be altered despite a higher actin content. This indicates again that the actin is aligned.

The accumulation of actin in a cortex of constant width indeed implies that the filament density must increase. We added a comment to the revised manuscript, in the paragraph in which we present the cortex width measurements: “Importantly, however, we did not detect a significant difference between the thickness of the equatorial cortex of metaphase and anaphase cells (Figure 2—figure supplement 1), indicating that equatorial actin accumulations in anaphase cells must be due to increased local filament density.” We do not think that the cortex width measurements provide direct information on the filament alignment state and hence prefer not to comment on alignment in this paragraph.

9) Subsection “Equatorial actin filament reorientation depends on myosin II activity”: In the context of myosin driven actin accumulation, the work of Wollrab et al. should be discussed here as they analysed in detail the recruitment and activity of myosin during ring contraction (Wollrab et al., 2016).

Following the suggestion, we reference this paper at the respective section: “This is consistent with small-scale organization within the actomyosin ring, where pronounced myosin II clusters also do not locally enrich actin to the same extent.” We would like to clarify that we do not claim that myosin drives actin accumulation (as the reviewer implies in the comment), but rather suggest that myosin II mediates reorientation of actin filaments within the equatorial cell cortex.

10) Subsection “Equatorial actin filaments reorient on planar cortex regions”: Important, now the authors stated that they detect alignment together with the onset of furrow ingression. It would be good to point out that the improved spatial resolution increased the sensitivity for changes in filament alignment and in contrast to the earlier experiments where they were not able to detect any changes in alignment at such early time points.

This was formulated imprecisely, as the anisotropy in this experiment does not increase earlier compared to the other experiments performed at central optical slices. Figure 3 shows that actin accumulates at the equator at ~90 s after anaphase onset, whereas the normalized emission ration increases only at ~120 s after anaphase onset (Figure 3; comparing post-anaphase onset frames to the metaphase interval -100 to 0 s and considering the relatively high s.d.). We rephrased the text for clarification.

11) Subsection “Cortical tension is highly directional at the cell equator”: How did the authors look at cortical flows? What is their definition of long vs short range?

We investigated cortical flows by kymographs and used the term long-range to refer to flow outside of the central region within about 5 μm of the cell equator. Figure 4 in our original manuscript showed only a small equatorial region, and it was hence not clear what we meant by long-range flow. For better documentation, we provide a new kymograph with a longer profile (new Figure 4) and omit the term “long-range”, as the revised figure directly visualizes the limited flow outside of the central regions.

12) Please see figure comments regarding the micro–surgery experiments.

We addressed all comments on the laser cutting experiments as explained below in the section on figures.

13) Discussion section: The discussion is very hand waving and needs thorough rewriting. It seems that out of nothing, the authors say now that everything is driven by local myosin activity. This is well possible, but then the observed network alignment is not a big deal, and the question remains how the myosin activity is localized to the ring. The authors fail to put their work in context with earlier work, such as Fishkind and Wang, how the experimental improvements help in understanding the network organization during cytokinesis, and how their findings will help to solve the ongoing debate about the most conclusive model.

We completely rewrote the discussion to clarify that cleavage furrow ingression is driven by local actomyosin contractility *at the cell equator*. The revised and extended Discussion section now explains in detail how our work relates to prior studies, in particular the study of Fishkind and Wang. We also discuss in detail our technical improvements.

14) “Our study shows that disordered actomyosin networks can generate sufficient mechanical tension to initiate cleavage furrow ingression in vertebrate cells”: The authors do not test whether the disordered actomyosin network generates enough tension for furrow ingression. Especially the blebbistatin experiments show that there are signals fostering actin filament polymerization at the furrow site without ingression. To support the above statement, one would have to do a laser ablation next to the developing furrow ingression and observe, if and how the ingression evolves further or not.

Our data show that the initiation of a cleavage furrow occurs in the absence of detectable actin filament alignment along the cell equator. We agree that it is suboptimal to include a statement on tension in the same sentence, and we have therefore removed this part of the sentence to separately discuss it in subsequent paragraphs. In the revised manuscript, we state: “Our study shows that the initial phase of cleavage furrow ingression is mediated by equatorial contraction of a disordered actomyosin network…”

15) “This suggests that vertebrate cells align the actomyosin networkdirectly at the site of highest contractility, whereby lateral cortical movements adjacent to the cell equator”: It's not very clear what the authors want to point out here. Cells do not align the actomyosin at the site of contractility, contractility itself aligns generally the actin filaments.

We thank the reviewer for indicating that this sentence was not very clear. We rephrased it to: “This suggests that during human cell cytokinesis, the actomyosin network aligns directly at the site of highest contractility, whereby lateral cortical movements adjacent to the cell equator might reflect drag imposed by highest network contractility at the equator.”

16) “Given that actomyosin rings mediate various other biological processes, including cellular wound healing and developmental morphogenesis, it will be interesting to further investigate the intricate relationship between contractile forces and actin network organization”: Very generic statement. Can be left out.

We prefer to keep this statement to point a broad readership towards potential implications of our work outside the cytokinesis field.

17) Major point 1: Interpretation of the low anisotropy regions in the mid–planeThe metaphase cell in Figure 1 shows a clear trend that the regions with strong signals in the average image show lower color saturation in the orientation map (white), hence, lower polarisation. This backs up my arguments about the scaling effect (major point 2 in my previous comments), i.e., denser the structure, lower the calculated anisotropy. Apart from the theory, the authors should provide good reasoning for the interpretation of "white" signal at the furrow as perpendicular alignment while those in the metaphase cell are clearly not representing such alignment.

We thank the reviewer for requesting clarification about a potential correlation between mean fluorescence intensity and the calculated polarization factor. Figure 1 shows a visualization of two parameters – mean fluorescence and polarization – in a single image, which is difficult to interpret quantitatively. In the revised manuscript, we hence provide an additional display of the same data in separate image panels for average fluorescence intensity and polarization, respectively (new Figure 1—figure supplement 1). These images clearly show that at early anaphase there is no correlation between fluorescence intensity and the calculated anisotropy, validating that the reduction of polarization measured at later stages of furrow ingression indeed indicates reorientation of actin filaments.

18) Major point 2: Comparison with a previous reportThe polarization microscope setups used in this study are most straight–forward and sensitive in detection of the polarization within the X–Y plane. Indeed, the pioneering work with a similar setup by Fishkind and Wang focused on this. Honestly, I don't understand why the authors insist on not taking this approach (e.g. flatten a cell in a way similar to Figure 3) on their LC–Polscope. This should allow straight–forward visualization of the filament alignment and direct comparison with the results by Fishkind and Wang. Anyway, Figure 3 data, in principle, provide a live version of Fishkind and Wang, which is indeed a drastic improvement. Similarities, differences and improvements should be properly discussed in Discussion, including the difference in the data analyses (simple ratiometry in Figure 3 vs (F_parallel – F_perpendicular)/(F_parallel + F_perpendicular in Fishkind and Wang).

We thank the reviewer for pointing out that we should point out more clearly how our method compares to the prior study. We have extended the Discussion section accordingly:

“The quantification of actin filament order at defined time points of cytokinesis has been established through several technical innovations. […]Yet, in combination with the fixed cell data, these experiments establish a detailed kinetic profile of actin network organization during cytokinesis progression.”

19) Major point 3: Laser surgery in Figure 4As I pointed out as Major point #10 in my previous comments, the comparison between "perpendicular" vs "parallel" cuts is not fair. Why a smaller round hole was not tried or is not suitable should be explained.

When implementing the laser microsurgery assays, we tried many different cutting geometries. We found that only experiments with thin line geometries yielded interpretable results: when scanning 2D area geometries with the pulsed high-energy laser, we either completely exploded the cells or did not induce measurable displacement of the cortex.

To further address the concern of the reviewer, we have included new data from cells that were cut along a linear path oriented at 45 degrees relative to the cleavage furrow (new Figure 4—figure supplement 1). These experiments show a diagonal cortical displacement, whereby the most pronounced displacement was oriented along the direction of the cell equator. The new data substantiate that the cortical tension is highest along the direction of the cell equator.

20) Subsection “Contraction of a randomly oriented actin network initiates cleavage furrow ingression”.As a sample with isotropic fluorescence dipole distribution, we imaged fluorescent plastic (Figure 1). We then normalized the polarization factor to be in a range between 1 and 0 for maximally aligned fluorophores (stress fibres) and isotropic plastic, respectively.I expect the isotropic plastic to have a polarisation factor of 0.5 much akin to the metaphase cortex. Is there something specific about the substrate?

In fluorescent plastic, the fluorophore dipoles are randomly oriented in all three spatial dimensions, which we use as a reference for a completely isotropic sample. The cell cortex of mitotic HeLa cells is relatively thin (~200 nm), which might confine actin filaments into a flat network. If the actin network contains filaments oriented along the cell surface, but with random orientations within the surface plane, this should yield a polarization factor of 0.5.

We rephrased the main text for clarification:

“The actin cortex underlying the plasma membrane of mitotic cells is relatively thin, which might confine actin filaments into a flat network. […] As each camera image pixel records signal from many fluorophores, this should yield a normalized polarization factor close to 0.5 along the metaphase cell cortex.”

21) Figure 1.During late stages of anaphase, the normalised polarisation at the poles appear to be more than 0.6 (mean) in Figure 1 and less than 0.6 in Figure 1. Can the authors explain the discrepancy?

The reason for the (very small) difference in the mean polarisation at the poles is that the initial dataset of 22 cells (as shown in Figure 1) was extended by 17 additional cells for a more accurate estimation of the mean as shown in Figure 1. The sample size of 22 for Figure 1 is indicated in the legend.

22) Figure 1.The authors seem to let go of a major point by not comparing the normalised potential of either equator or poles for different time points. This would have made a nice point regarding orientation of actin over time.

This might be interesting, yet the increase in normalized polarization factor at the poles is small. We now comment on this in the revised main text:

“We also noticed a small increase of normalized polarization factor at the poles during late stages of cytokinesis, which might reflect cortical reorganization when cells reattach to the glass substratum during mitotic exit.”

23) Laser Microsurgery experiment:This is indeed a very interesting observation. The cut in the parallel direction on the cell cortex doesn't induce cortex movement which is in stark contrast to a similar cut made in the perpendicular orientation. It will however be interesting to see whether the parallel slit can induce outward movement if its horizontal dimension (which is a line currently) is increased. If so, then does this minimum lateral dimension (when outward motion is achieved) correspond to the FWHM of the anisotropic actin network measured in the paper (4.4.um)?

When we developed the laser microsurgery assays, we tried many different geometries and found that exposing broader 2D areas to the pulsed high-energy laser leads to immediate explosion of cells, whereby the edge of the cut cortex cannot be tracked. We therefore can only use thin lines for the laser microsurgery experiments.

To address the reviewer’s question about alternative cutting geometries, we added new data with a cutting line that is oriented at 45 degrees relative to the cleavage furrow (Figure 4—figure supplement 1). This shows a diagonal cortical displacement following the orientation of the cell equator. These new data substantiate that the tension is distributed aniostropically whereby the peak tension is aligned with the cell equator.

24) As a general interest it would be interesting to know, what happens to these cells after the cut is made. Do these cells show a healing response or abort cytokinesis and undergo cell death.

We provide this information in the revised main text: “Time-lapse microscopy showed that following the rapid cortical displacement after laser cutting, most cells repaired the cell cortex and resumed furrow ingression to completion (8 out of 12 cells within 10 min; 2 cells stopped furrow ingression and 2 cells died).”

Issues with figuresFigure 1: The question marks can be left out.

We removed the question marks as suggested.

Figure 1) The authors might want to discuss why the cortical actin shows parallel alignment along the plasma membrane due to the overall confinement of the actin filaments along the membrane. And that the white area along the furrow indicates the alignment of filaments along the z axis. Since this kind of measurements are not common for the major audience, a careful description and discussion are important.

We now discuss the geometrical considerations and data shown in Figure 1 in more detail:

“The canonical “purse-string” model of cytokinesis proposes that actin filaments align along the cell equator to drive furrow ingression by anti-parallel sliding (Figure 1). […] At late stages of cytokinesis, we indeed observed a pronounced reduction of the normalized polarization factor at the cell equator (Figure 1, 201-300 s, cortex area with low color saturation; quantification in 1F), consistent with preferential orientation of actin filaments along the cell equator.”

Figure 1) Given the spread of the data, tracing the difference of individual cells would be helpful by connecting the corresponding red and blue dots.

We thank for the suggestion how to improve data visualization. Given the large number of cells, however, we found that connecting pairs of dots does not help with the data interpretation. To give the reader additional information on the accuracy of the data, we added bars indicating s.e.m. to the revised Figure 1.

Figure 2: The cell cortex appears jagged and the 0 and 90 degree channels do not seem always properly aligned (very clear at 330sec). Why is that the case? Channel alignment is essential for the polarization analysis.

The slight misalignment between some cortical regions in the two image channels is caused by sequential image acquisition, while the cleavage furrow moves in live cells. In our quantitative analysis, we compensate for this by using line widths that contain all signal from both channels. We now comment on this in the revised Materials and methods section: “This segmentation contour line had a width that completely contained the cortex signal even when the two image channels were slightly shifted owing to cleave furrow ingression during their sequential acquisition (e.g., Figure 2, control, 290 s).”

Figure 2—figure supplement 1: It would be instructive if the authors could first show the raw intensities before computing the ratio.

Without the normalization of individual data points through calculating H/V ratios, the raw intensities for individual channels appear very noisy and do not convey meaningful information, in our view. We have included the plot of raw intensities below but do not think it is helpful to incorporate this into the manuscript.

Figure 2—figure supplement 1:How can you obtain negative values for the actin cortex thickness?

The negative values represent measurement noise and based on the reviewer’s comment we realize that it would be more accurate to re-label the Y-axis in this plot. For this figure, we calculated the difference between the center positions of Gaussian functions fitted to the MyrPalm and SiR-actin channels, respectively. We used line profiles oriented perpendicularly to the cell cortex, drawn from the inside towards the outside of cells. This yields positive values when the SiR-actin center position lies on the cytoplasmic face relative to the MyrPalm position. The negative values result from line profiles where the center position of SiR-actin was determined to lie on the extracellular face relative to the MyrPalm position, most likely reflecting measurement noise. We clarified this by rephrasing the axis label to “Radial displacement (MyrPalm – SiR-actin) (nm)”.

Figure 3: The plot indicates that changes in network orientation appear at about 100sec. this would mean that alignment would go hand in hand with the furrow ingression. The authors should discuss this point as it is in conflict with the earlier conclusion that alignment can be only detected at later time points.

We disagree with the reviewer’s interpretation of the data shown in this plot. Compared to the metaphase interval (-100 to 0 s), this plot does not show an increase in normalized emission ratio until ~120 s post anaphase onset (considering the variability indicated by the shaded areas, indicating s.d.). We clarified this in the main text.

Figure 4: The authors should show a bleach control to show that their ablation protocol really cuts the actin network. They also should provide data to indicate whether the plasma membrane stays intact or not during the procedure.

We have included a bleach control, showing that photobleaching of Actin-EGFP with a conventional 488 nm laser results in homogeneous recovery in the bleached region without lateral cortical displacement (new Figure 4—figure supplement 1). Hence, the lateral displacement is a specific response to cutting with the pulsed laser.

We have also added new data showing that the pulsed laser also cuts the plasma membrane, as imaged by Myr-Palm-EGFP (new Figure 4—figure supplement 1).

In addition, it would be very instructive for the authors conclusion to perform a laser ablation of the network adjacent to the cytokinetic ring in the beginning of furrow ingression and to observe whether the contraction progresses or not. If the global contraction of the random actin network is necessary for the cytokinetic ring, then this ablation should perturb the process.

The laser microsurgery does not consistently cause persistent damage of the actomyosin ring, as most cells resumed cytokinesis some time after laser microsurgery, even when the central region of the actomyosin ring was destroyed. We provide these information in the revised main text: “Timelapse microscopy further showed that following the rapid cortical displacement after laser cutting, most cells repaired the cell cortex and resumed furrow ingression to completion (8 out of 12 cells within 10 min; 2 cells stopped furrow ingression and 2 cells died).” In light of these observations, we do not think that the suggested experiment will be informative. However, we have added new laser microsurgery data with a diagonal cutting path, which provides further insight into the distribution of cortical tension (new Figure 4—figure supplement 1).

Figure 5: The schematic looks nice, but is not very instructive. In addition, the model is not properly formulated (i.e. in a theoretical way) and the claims are not presented in clear connection to the presented data. How is the equatorial contractility generated in the first place (–> increased local myosin activity)? how to explain lateral cortex displacement without increase of tension (–> less connectivity in the network? Shorter filaments?) How is the higher tension along the equator generated (–>via filament alignment)?

We revised the model figure and rephrased the Discussion section for clarification: “In this model, the activation of myosin II at the cell equator would induce local contractility in a cortical network of randomly oriented actin filaments. Along the cell equator, each given point experiences counteracting contractile forces from neighboring regions, which would build up high levels of mechanical tension along the circumference of the cell. In the direction perpendicular to the cell equator, highly contractile equatorial cortex regions connect to neighboring regions of lower contractility, which would move towards the equator without building high levels of tension. The resulting asymmetric network deformation could explain the filament reorientation towards the direction of the cell equator, which might in turn further increase the contractile forces oriented along this direction.”

Figure 1—figure supplement 1.The area for selected at the poles is mentioned as 16um in the figure against 10um in the text (Materials and methods section).

We used 16 μm line paths for the analyses of polar regions. We corrected the Materials and methods section accordingly.

Figure 2Curvature change in Figure 2 appears to reduce post 350 second. No such trend however is observed for Figure 2 for essentially the same measurement.

Figure 2 shows an example curve from a single cell, whereas Figure 2 shows the mean and s.d. of the entire dataset of 18 cells. The shaded gray area in Figure 2 (indicating s.d.) reveals a high variability in the curvature at late furrow ingression stages (>250 s), which is largely due to the variable geometry by which the two daughter cells organize relative to each other. Some sister cells move apart and then elongate their intercellular bridge, resulting in a reduction of curvature (as for the cell shown in Figure 2). However, this variability does not affect on our analysis of SiR-actin polarization, as we only consider early stages of furrow ingression (indicated by green areas).

Figure 2 and Figure 1.In Figure 2, there appears to be no net change in normalised emission ratio, indicative of polarisation, at the poles. However, in Figure 1, there seems to be a modest increase in normalised polarisation.

These differences might be due to the different F-actin staining methods used in the two experiments (Alexa Fluor 488-phalloidin versus SiR-actin) or the different microscope set-up. As the differences in polar actin network anisotropy are only minor, we do not think that this compromises the interpretation of our data at the cleavage furrow, which is consistent in both types of experiments.

Figure 4.It would be interesting to know whether the increasing movement of the network on ablation scales with increased actin alignment.

The increase in initial outward movement indeed correlates with an increase in normalized emission ratio. We now show this in a new Figure 4 panel J.